# PowerBacGWAS: a computational pipeline to perform power calculations for bacterial genome-wide association studies

Francesc Coll [1✉], Theodore Gouliouris [2,3], Sebastian Bruchmann [4], Jody Phelan [1], Kathy E. Raven[2], Taane G. Clark [1,5], Julian Parkhill [4] & Sharon J. Peacock [2]

Genome-wide association studies (GWAS) are increasingly being applied to investigate the genetic basis of bacterial traits. However, approaches to perform power calculations for bacterial GWAS are limited. Here we implemented two alternative approaches to conduct power calculations using existing collections of bacterial genomes. First, a sub-sampling approach was undertaken to reduce the allele frequency and effect size of a known and detectable genotype-phenotype relationship by modifying phenotype labels. Second, a phenotype-simulation approach was conducted to simulate phenotypes from existing genetic variants. We implemented both approaches into a computational pipeline (PowerBacGWAS) that supports power calculations for burden testing, pan-genome and variant GWAS; and applied it to collections of *Enterococcus faecium*, *Klebsiella pneumoniae* and *Mycobacterium tuberculosis*. We used this pipeline to determine sample sizes required to detect causal variants of different minor allele frequencies (MAF), effect sizes and phenotype heritability, and studied the effect of homoplasy and population diversity on the power to detect causal variants. Our pipeline and user documentation are made available and can be applied to other bacterial populations. PowerBacGWAS can be used to determine sample sizes required to find statistically significant associations, or the associations detectable with a given sample size. We recommend to perform power calculations using existing genomes of the bacterial species and population of study.

[1] Department of Infection Biology, Faculty of Infectious & Tropical Diseases, London School of Hygiene & Tropical Medicine, London, UK. [2] Department of Medicine, University of Cambridge, Cambridge, UK. [3] Cambridge University Hospitals NHS Foundation Trust, Cambridge, UK. [4] Department of Veterinary Medicine, University of Cambridge, Cambridge, UK. [5] Faculty of Epidemiology and Population Health, Department of Infectious Disease Epidemiology, London School of Hygiene & Tropical Medicine, London, UK. ✉email: Francesc.Coll@lshtm.ac.uk

Bacterial genome-wide association studies (GWAS) are a group of comparative genomics techniques aimed at identifying genetic variants in bacterial genomes that correlate with a phenotypic trait that is variable in a population. The introduction of bacterial GWAS became possible as a result of the increase in the number of whole-genome sequenced isolates. In the last few years, bacterial GWAS have been applied to study the genetic basis of a range of bacterial traits including antibiotic susceptibility[1–4], susceptibility to disinfectants[5], host specificity[6,7], transmissibility[8], carriage duration[9], adaptation to humans[10], disease presentation or clinical infectious disease phenotypes[11–13], invasiveness[14–19] and disease severity and outcomes[20]. See refs. [6,21–23]. for reviews on the topic.

The wider adoption of bacterial GWAS is becoming possible thanks to the development of specialised tools tailored to the peculiarities of bacterial genomes[1,24,25], adapted from tools originally developed for human GWAS. The use of units of association that capture genetic variation in both the core and accessory genomes was an important development for bacterial GWAS; including the use of k-mers[26], presence or absence of genes in the accessory genome[25], and unitigs[27]. Methodological advances include the use of linear mixed models (LMMs), which better control for fine population structure, and the deconvolution of lineage and locus effects[1]. Phylogenetic approaches that test for the co-evolution of genetic variants and traits have also been developed[28,29].

A prerequisite for a successful GWAS is to perform power calculations to ensure that there are sufficient numbers of strains in the study to find statistically significant associations, or alternatively to determine the effect sizes detectable with a given collection of isolates. Variables affecting the power to detect causal variants include those related to the causal variants per se, such as their minor allele frequency (MAF), effect size or how they are distributed across the phylogeny (e.g. degree of homoplasy); and those related to the bacterial population under study, such as the degree of population diversity and structuring. Approaches developed for power calculations in human GWAS[30] cannot be applied to bacteria due to the unique characteristics of bacterial populations (clonal reproduction, strong population structure and uneven and varying degrees of recombination).

In this work, we show how existing collections of bacterial genomes can be harnessed to conduct power calculations for bacterial GWAS and measure the effect of these variables on the ability to discover causal variants. We implemented two new approaches to perform power calculations (called Power-BacGWAS) and here apply it to three different bacterial species of clinical importance: *Enterococcus faecium* (frequent cause of nosocomial infections), *Klebsiella pneumoniae* (a cause of bacterial pneumonia) and *Mycobacterium tuberculosis* (causes tuberculosis). The methodology is general and compatible for a range of genomic variation including point mutations, indels and variation in gene content. We have made the code public (https://github.com/francesccoll/powerbacgwas) and provide user documentation on how to apply it to other bacterial populations.

## Results

### Implementation of GWAS power calculations for bacterial populations.
The successful identification of phenotype-genotype associations in bacteria is influenced by multiple factors which vary strongly between bacterial species such as population structure, gene presence/absence, homoplasy and phenotype heritability. It is therefore not possible to create common statistical methods of estimating power in GWAS studies such as those used in well-studied human populations. To estimate the sample sizes required to detect phenotype–genotype associations in bacteria we

implemented two new approaches based directly on real bacterial genomes and phylogenies (Fig. 1). In the first approach (sub-sampling approach), we take a known genotype-phenotype relationship and sequentially sub-sample the population to reduce sample sizes, allele frequency (AF) and effect sizes, to discover at which sub-sample sizes we can still recover the known associations. In the second (phenotype-simulation approach), phenotypes are simulated from randomly selected genetic variants meeting a range of parameters: minor allele frequency (MAF), effect size and sample size. We then perform GWAS to identify the sample size needed to recover the simulated genotype–phenotype relationship. Both approaches require the use of an existing collection of whole-genome sequenced strains of the same species (Fig. 1). The sub-sampling approach additionally requires a measured phenotype and list of known causal variants. In any case, bacterial genomes are not simulated or modified; only phenotype strains' labels are changed or simulated to achieve the desired combination of parameters.

### Selection of strain collections and known genotype–phenotype relationships.
Table 1 shows the strain collections used in this study. These collections were assembled from published datasets of whole-genome sequenced strains to include a representative of a gram-positive organism (*E. faecium*), a gram-negative (*K. pneumoniae*) and a species of limited genetic diversity (*M. tuberculosis*). To test the effect of strain diversity in a population on the power of detecting causal variants, we assembled a population representative of the species' overall strain diversity (hereafter referred to as species-wide population), and a second population of lower genetic diversity, made up of samples from a single clade (single-clade population). Table 1 summarises the overall genetic diversity of each population in terms of pan-genome size, number of SNP sites and average pairwise genetic distance.

Next, we searched for a known antimicrobial resistance (AMR) phenotype–genotype relationship in each population that could be used to perform power calculations (See Methods section for rationale and selection criteria of AMR phenotypes). For the species-wide *E. faecium* collection ($n = 1432$), we could use kanamycin resistance (35.3% resistant, 23.3% susceptible, 41.4% not tested) caused by the aminoglycoside resistance *aph(3′)-IIIa* gene[31] (AF = 56.3%, odds ratio (OR) = 1083); and streptomycin resistance for the single-clade clade A1 population ($n = 761$, 34.5% resistant, 60.3% susceptible and 5.2% no tested) determined by the streptomycin-resistance *ant(6′)-Ia* gene[31] (AF = 34%, OR = 8986) (Table 1). For *K. pneumoniae*, we used meropenem resistance (21% resistant, 69.1% susceptible, 9.9% not tested) caused mainly by $bla_{KPC}$ carbapenamese (AF = 12%, OR = 180) for the species-wide population ($n = 2628$), but could not use it for the single-clade ST288 population ($n = 1193$) due to the unbalanced proportion of resistant (95.4%) and susceptible cases (1.3%). Isoniazid resistance was used for both *M. tuberculosis* populations ($n = 2655$, AF = 20%, OR = 220; $n = 1139$, AF = 13%, OR = 166), which is determined by well-known *katG* mutations[32,33].

### Application and interpretation of power calculations: sub-sampling approach.
Figure 2 and Supplementary Table 1 show the sample sizes required to detect known AMR genotype–phenotype relationships with 80% power as obtained by the sub-sampling approach. These results show that, as expected, the larger the effect size and AF of causal variants, the smaller the sample sizes required to detect them using a GWAS. The pan-genome GWAS, conducted for *E. faecium* and *K. pneumoniae* populations to detect acquired AMR genes, yielded very comparable results, both between populations of the same species and across species. Specifically, a sample size of 500 to 700 genomes was enough to detect moderate (OR = 5) to very large (OR = 100) effect sizes of genes present in at least 10% of

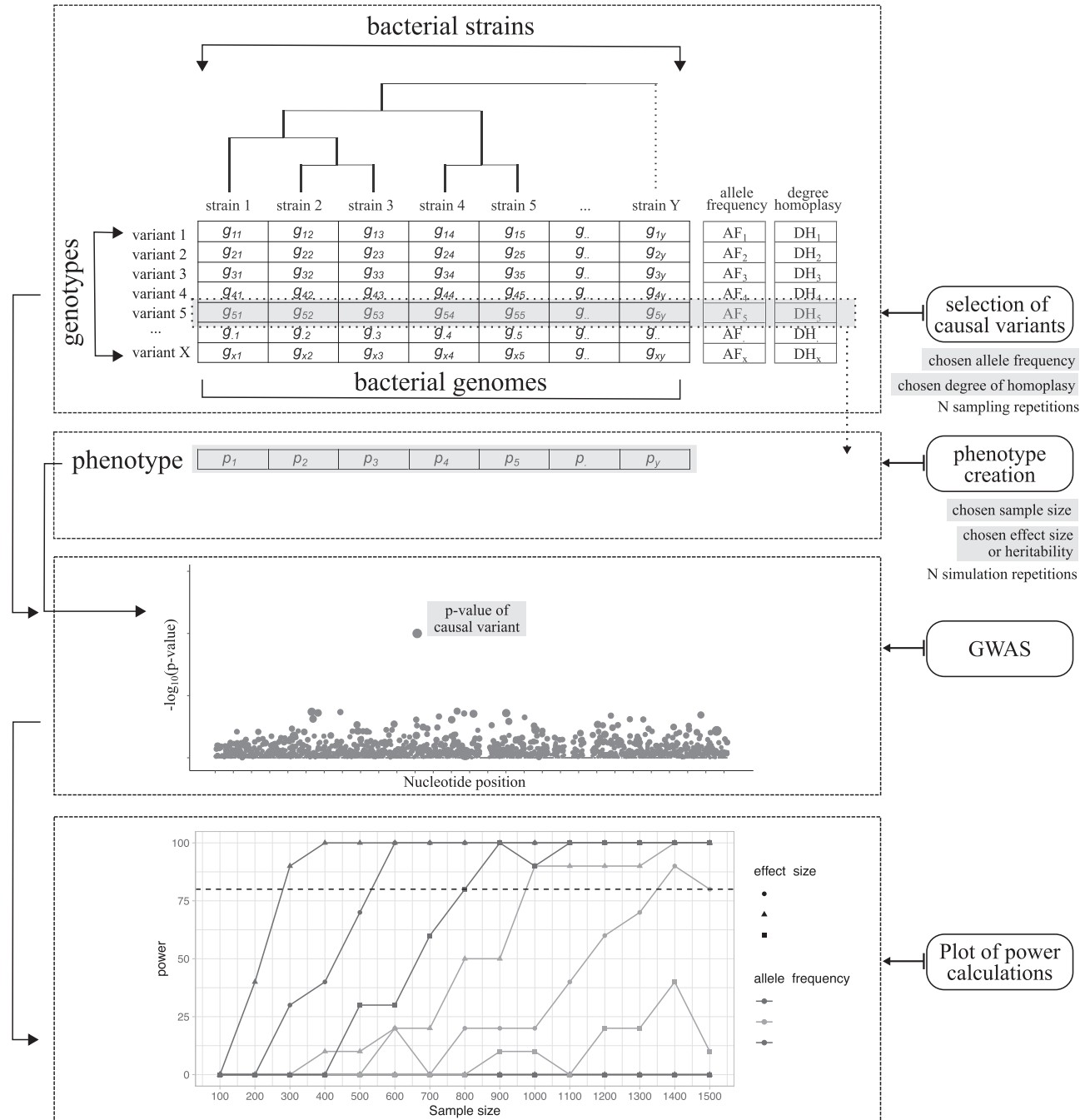

**Fig. 1 Approach to bacterial GWAS power calculations.** Four steps were implemented to conduct power calculations. First, known or randomly sampled causal variants are chosen from existing genotypes, in the sub-sampling or phenotype simulation approach, respectively. In the latter, causal variants meeting a range of selected MAF and degree of homoplasy are selected. Second, phenotypes are either modified from existing ones (sub-sampling approach) or simulated from randomly selected genotypes (phenotype simulation approach) to achieve the range of chosen sample sizes and effect sizes (or heritability values). Third, a genome-wide association study (GWAS) is conducted for each combination of parameters and p-values of causal variant extracted. And forth, power is calculated as the proportion of GWAS replicates in which the causal variant is above the Bonferroni-corrected genome-wide significance threshold.

the population (Supplementary Table 1); or 400–600 for causal genes of very large effect sizes (OR = 100) at 5% frequency, depending on the bacterial population studied. Larger sample sizes of at least 800–1100 genomes were needed to detect genes of moderate effect sizes at 5% frequency. Genes of small effect sizes (OR = 1.5) could not be detected using the maximum sample sizes available in our collections. Power calculations for *M. tuberculosis* showed that the bacterial population had an effect on the power to detect causal mutated genes in a burden GWAS: lower sample sizes were required

for the population with lower diversity (single-clade) compared to the species-wide population, to detect genes of the same MAF and effect sizes (Supplementary Table 1).

**Application and interpretation of power calculations: phenotype simulation approach**. The phenotype simulation approach allowed us to test the effect of a wider range of parameters on the power to detect acquired genes (pan-genome GWAS), individual

**Table 1 Bacterial species, strain collections and antibiotic susceptibility phenotypes used in this study.**

| Bacterial species | Strain collection | # of isolates (diversity) | LD: median $R^2$ (IQ range) | # of SNP sites | # of genes in pan-genome | AMR phenotype (% R and S)[a] | AMR causal variants | AMR causal variants: AF[b], OR and GWAS p-value |
|---|---|---|---|---|---|---|---|---|
| Enterococcus faecium | Species-wide | n = 1432 (5.6 SNPs/kb) | 0.65 (0.37–0.95) | 263,875 | 11,800 | Kanamycin susceptibility (35.3%, 23.3%) | aph(3')-IIIa | AF: 56.3% OR: 1083 p-value: $8.25 \times 10^{-145}$ |
| | Single-clade | n = 761 (2.4 SNPs/kb) | 0.50 (0.28–0.98) | 50,790 | 5443 | Streptomycin susceptibility (34.5%, 60.3%) | ant(6)-Ia/aad(6) | AF: 34% OR: 8986 p-value: $1.61 \times 10^{-51}$ |
| Klebsiella pneumoniae | Species-wide | n = 2628 (5.2 SNPs/kb) | 0.67 (0.37–1.00) | 543,165 | 30,772 | Meropenem susceptibility (21%, 69.1%) | $bla_{KPC}$ | AF: 12% OR: 180 p-value: $8.90 \times 10^{-110}$ |
| | Single-clade | n = 1193 (0.11 SNPs/kb) | 0.78 (0.50–0.96) | 46,541 | 23,708 | Meropenem susceptibility (95.4%, 1.3%) | $bla_{KPC}$ | AF: 72% OR: NA[c] p-value: NA[c] |
| Mycobacterium tuberculosis | Species-wide | n = 2655 (0.2 SNPs/kb) | 0.86 (0.39–1.00) | 93,995 | 21,678 | Isoniazid susceptibility (30.9%, 66.4%) | nsSNPs in katG | AF: 20% OR: 220 p-value: $2.54 \times 10^{-101}$ |
| | Single-clade[e] | n = 1139 (0.05 SNPs/kb) | 0.98 (0.40–1.00) | 24,467 | 10,130 | Isoniazid susceptibility (23.8%, 71.7%) | nsSNPs in katG | AF: 13% OR: 166 p-value: $6.40 \times 10^{-73}$ |

Summary table of strain collections used in this study. The average diversity (third column) was calculated as the mean pairwise genetic distance between isolates, expressed as number of SNPs per kilobase. The number of SNP sites in the chromosome (forth column; extracted from the VCF file) and number of genes in the pan-genome (fifth column; extracted from Panaroo's output), both calculated across all isolates, indicate the degree of diversity within each collection. The last columns show the AMR phenotypes and causal variants used by the sub-sampling approach to perform power calculations. The single-clade collections correspond to: clade A1 isolates for E. faecium; CC258 isolates for K. pneumoniae; and lineage 4.3 isolates for Mycobacterium tuberculosis.
[a]The percentage of resistant and susceptible isolates may not amount to 100%, as a subset of isolates were not tested.
[b]The MAF was calculated in the whole population not in just the samples phenotyped for the antibiotic in question.
[c]The unbalanced number of cases and controls prevented running GWAS.
SNPs/kb Single Nucleotide Polymorphisms per kilobase, AF allele frequency, OR odds ratio, nsSNP non-synonymous SNPs, LD linkage disequilibrium.

**a** Pan-genome GWAS power calculations in *E. faecium*
using *aph(3')-IIIa* kanamycin-conferring gene and kanamycin susceptibility phenotypes

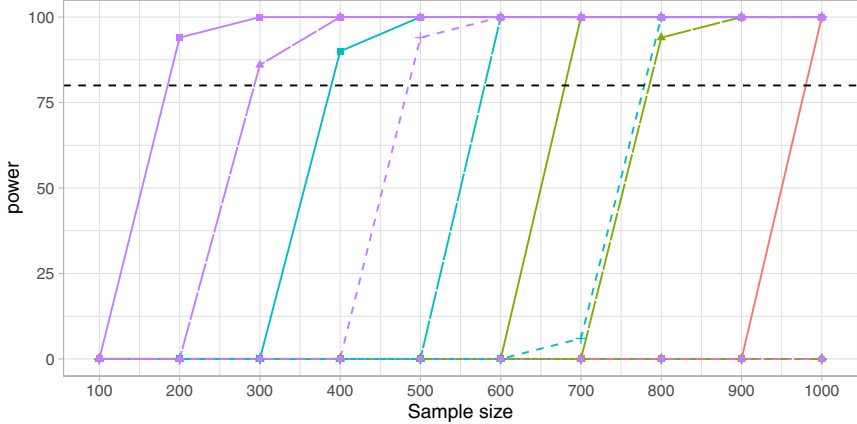

**b** Pan-genome GWAS power calculations in *K. pneumoniae*
using *bla$_{KPC}$* carbapenem-conferring gene and meropenem susceptibility phenotypes

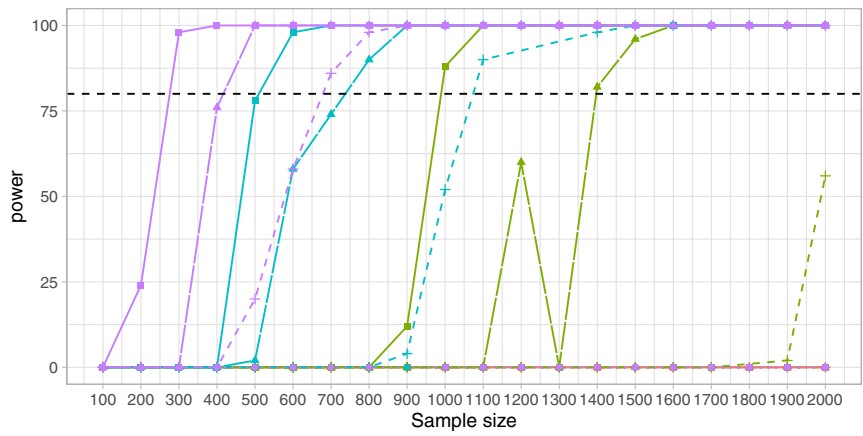

**c** Burden testing GWAS power calculations in *M. tuberculosis*
using *katG* isoniazid-conferring mutations and isoniazid susceptibility phenotypes

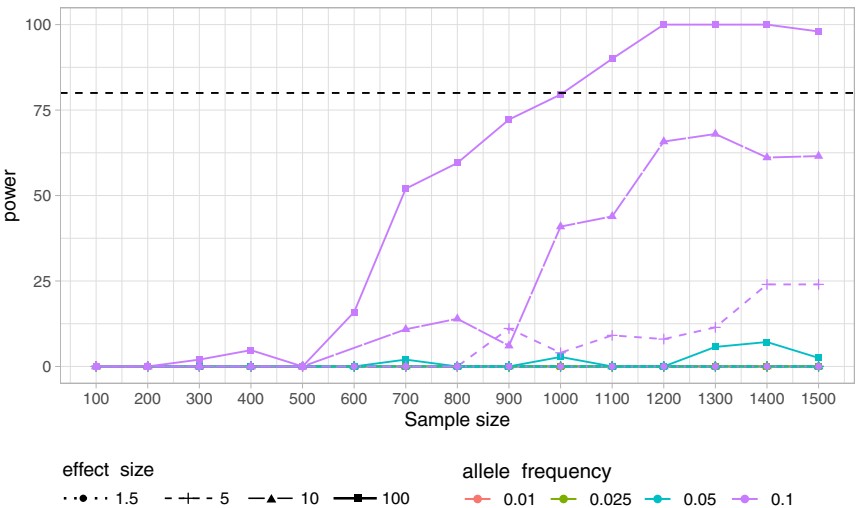

SNPs (variant GWAS) and mutated genes (burden testing GWAS) causing binary phenotypes. Table 2 and Supplementary Fig. 1 show the sample sizes required to detect causal acquired genes with 80% power in a pan-genome GWAS. These sample sizes are comparable to those obtained by the sub-sampling approach for the detection of acquired AMR genes (Supplementary Table 1). Specifically, a minimum of 500–600 genomes

were needed to detect moderate (OR = 5) to very large (OR = 100) effect sizes of genes of 10% frequency, regardless of the bacterial population considered (Table 2); and 200–500 or less to detect genes of 25% frequency. Larger sample sizes of at least 2000–2500 genomes are needed to detect genes of 2.5% frequency. In addition, we tested the effect of increasing heritability and found that, as expected, the higher the heritability of causal

**Fig. 2 Power calculations obtained using the sub-sampling approach for the detection of AMR genes.** Results of running GWAS power calculations applying the sub-sampling approach for the detection of known AMR genotype-phenotype relationships (binary phenotype). These plots show the sample sizes required to detect AMR causal genes of different AF and effect sizes (for which full heritability is assumed). The y-axis shows the power, calculated as the proportion of GWAS replicates in which the causal AMR gene is above the Bonferroni-corrected genome-wide significance threshold. The black and dotted horizontal line marks 80% power. Sample sizes are represented in the x-axis. The colour of lines denotes different AF whereas point shapes and line types effect sizes in odds ratio units. The power calculation results presented here are those for the species-wide populations, see Supplementary Table 1 for sample sizes required in both species-wide and single-clade populations. Pan-genome GWAS was run to detect acquired AMR genes in *E. faecium* (**a**) and *K. pneumoniae* (**b**) populations. A burden test GWAS was applied to *M. tuberculosis* (**c**).

**Table 2 Sample sizes required to detect causal genes of different MAF and effect sizes in a pan-genome GWAS.**

| Bacterial species | Strain collection | Gene frequency (%) | Effect size (odds ratio) | | | |
|---|---|---|---|---|---|---|
| | | | Small (1.5) | Moderate (5) | Large (10) | Very large (100) |
| *Enterococcus faecium* (pan-genome GWAS) | Species-wide (n = 1432) | 1 | – | – | – | – |
| | | 2.5 | – | – | – | 1100 |
| | | 5 | – | 1000 | 600 | 500 |
| | | 10 | – | 500 | 400 | 200 |
| | | 25 | – | 200 | 200 | 100 |
| | Single-clade (n = 1531) | 0–1 | – | – | – | – |
| | | 2.5 | – | – | 1400 | 1000 |
| | | 5 | – | – | – | – |
| | | 10 | – | 600 | 400 | 300 |
| | | 25 | – | 300 | 200 | 100 |
| *Klebsiella pneumoniae* (pan-genome GWAS) | Species-wide (n = 2628) | 1 | – | – | – | – |
| | | 2.5 | – | 2500 | 1600 | 1200 |
| | | 5 | – | 1500 | 1000 | 700 |
| | | 10 | – | 600 | 400 | 300 |
| | | 25 | – | 500 | 400 | 200 |
| | Single-clade (n = 1193) | 0–1 | – | – | – | – |
| | | 2.5 | – | – | – | 1000 |
| | | 5 | – | 900 | 700 | 500 |
| | | 10 | – | 500 | 300 | 200 |
| | | 25 | – | 300 | 200 | 100 |
| *Mycobacterium tuberculosis* (pan-genome GWAS) | Species-wide (n = 2655) | 1 | – | – | – | – |
| | | 2.5 | – | 2000 | 1300 | 1000 |
| | | 5 | – | 1100 | 700 | 500 |
| | | 10 | – | – | 900 | 500 |
| | | 25 | – | 300 | 200 | 100 |
| | Single-clade (n = 1139) | 0–1 | – | – | – | – |
| | | 2.5 | – | – | – | 1000 |
| | | 5 | – | 900 | 700 | 500 |
| | | 10 | – | 500 | 300 | 200 |
| | | 25 | – | 300 | 200 | 100 |

*MAF* minor allele frequency, - non-detectable with 80% power.
Results of running GWAS power calculations applying the phenotype simulation approach (binary phenotype, full heritability assumed). This table shows the minimum sample sizes required to detect acquired genes of different effect sizes (in odds ratio units) and gene frequencies in a pan-genome GWAS with 80% power, in both species-wide and single-clade populations.

genes, the lower the sample sizes required to detect them (Supplementary Table 2). Increasing heritability resulted in a sharp decrease in sample sizes required to detect common genes (i.e. 25% frequency), but had no, or little effect, on the detection of rarer genes (e.g. 2.5% frequency).

We next conducted power calculations for the detection of individual SNPs (SNP GWAS Supplementary Fig. 2) and mutated genes (burden GWAS, Supplementary Fig. 3). We found that burden testing had more power (i.e. required lower sample sizes to detect the same effect sizes) than a SNP GWAS and could detect mutated genes down to 2.5% MAF, not detectable by a SNP GWAS (Table 3). Here, the MAF of genes in a burden test refers to the percentage of samples carrying one or multiple SNPs in the same gene. The sample sizes required to detect SNPs of the same MAF and effect size (Table 3) with 80% power varied by population, although they were lower when using burden testing. Higher heritability resulted in more power to detect common

SNPs (Supplementary Table 3) but had little effect on the detection of rarer SNPs. Next, we studied the effect that the degree of homoplasy may have on the power to detect causal variants.

**Effect of homoplasy level**. The degree of homoplasy of causal SNPs, that is, the number of times SNPs arose independently in the phylogeny, had a big impact on the ability of GWAS to detect them (Fig. 3). GWAS were more powered at detecting highly homoplasic SNPs, although the effect was less pronounced for low-MAF SNPs (Supplementary Table 4). As an example, highly homoplasic SNPs (acquired 50 to 100 times) in *E. faecium* at 10% MAF could be detected with half the sample sizes needed to detect low homoplasic SNPs (acquired 1–5 times) of the same MAF, regardless of their effect size, a pattern we observed in all bacterial populations studied. We next studied the effect of the bacterial population on the power to detect SNPs. In

**Table 3 Sample sizes required to detect SNPs and mutated genes of different MAF and effect sizes.**

| Bacterial species | Strain collection | MAF (%) | Variant GWAS Effect size (odds ratio) | | | | Burden GWAS | | | |
|---|---|---|---|---|---|---|---|---|---|---|
| | | | Small (1.5) | Moderate (5) | Large (10) | Very large (100) | Small (1.5) | Moderate (5) | Large (10) | Very large (100) |
| E. faecium | Species-wide (n = 1432) | 1 | - | - | - | - | - | - | - | - |
| | | 2.5 | - | - | - | - | - | - | - | 1000 |
| | | 5 | - | - | - | - | - | - | - | - |
| | | 10 | - | - | 1200 | 700 | - | 900 | 700 | 400 |
| | | 25 | - | 1200 | 500 | 400 | - | 900 | 400 | 200 |
| | Single-clade (n = 1531) | 1 | - | - | - | - | - | - | - | - |
| | | 2.5 | - | - | - | - | - | - | 1300 | 900 |
| | | 5 | - | - | - | - | - | - | 900 | 700 |
| | | 10 | - | 1200 | 800 | 500 | - | 1100 | 700 | 400 |
| | | 25 | - | 1100 | 700 | 400 | - | 600 | 300 | 200 |
| K. pneumoniae | Species-wide (n = 2628) | 1 | - | - | - | - | - | - | - | - |
| | | 2.5 | - | - | - | 2000 | - | 2000 | 1300 | 1000 |
| | | 5 | - | 2000 | 1200 | 800 | - | 1000 | 700 | 500 |
| | | 10 | - | 800 | 600 | 500 | - | 700 | 400 | 300 |
| | | 25 | - | 300 | 200 | 100 | - | 400 | 200 | 200 |
| | Single-clade (n = 1193) | 1 | - | - | - | - | - | - | - | - |
| | | 2.5 | - | - | - | - | - | - | - | 1100 |
| | | 5 | - | - | - | - | - | - | - | 800 |
| | | 10 | - | - | - | - | - | - | - | - |
| | | 25 | - | 900 | 600 | 300 | - | 900 | 700 | 400 |
| M. tuberculosis | Species-wide (n = 2655) | 1 | - | - | - | - | - | - | - | - |
| | | 2.5 | - | - | - | - | - | 2000 | 1400 | 1000 |
| | | 5 | - | - | - | - | - | 1400 | 1000 | 700 |
| | | 10 | - | - | - | - | - | 1500 | 1200 | 600 |
| | | 25 | - | 1300 | 800 | 500 | - | 900 | 500 | 200 |
| | Single-clade (n = 1139) | 0-1 | - | - | - | - | - | - | - | - |
| | | 2.5 | - | - | - | - | - | - | - | 1000 |
| | | 5 | - | - | - | - | - | - | 800 | 600 |
| | | 10 | - | 900 | 600 | 500 | - | - | 900 | 500 |
| | | 25 | NA | 400 | 200 | 100 | - | 600 | 300 | 200 |

MAF minor allele frequency, NA no variants available with that MAF, - non-detectable with 80% power.

Results of running GWAS power calculations applying the phenotype simulation approach (binary phenotype, full heritability assumed). This table shows the minimum sample sizes required to detect acquired variants (i.e. mutations in the bacterial chromosome) of different effect sizes (in odds ratio units) and MAF using a variant or burden test GWAS with 80% power, in both species-wide and single-clade populations. Supplementary Figs. 2 and 3 show the PowerBacGWAS plots from which the results in this table were extracted from MAF minor allele frequency, NA no variants available with that MAF, - non-detectable with 80% power.

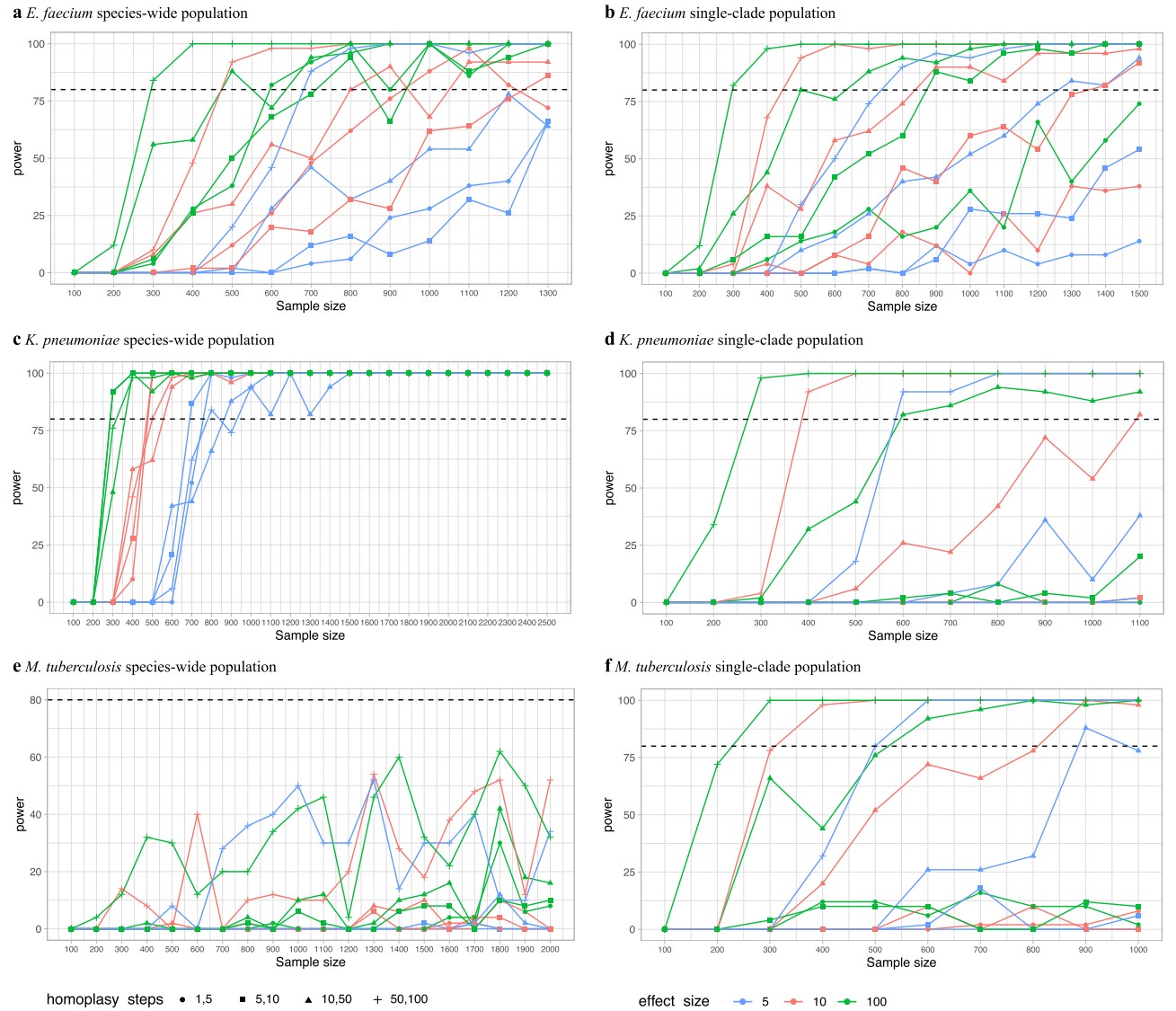

**Fig. 3 Effect of degree of homoplasy on the power to detect SNPs obtained using the phenotype-simulation approach.** These plots show the sample sizes required to detect causal SNPs of different effect sizes (in odds ratio units, showed as different colours) and degrees of homoplasy (number of independent acquisitions, shown as different point shapes) when simulating binary phenotypes (full heritability assumed). The power calculation results presented here are those for SNPs of 10% MAF, in both species-wide (panels **a**, **c**, **e**) and single-clade populations (panels **b**, **d**, **f**), see Supplementary Table 4 for SNPs of different MAF. The power in Fig. 3e, i.e. for SNPs with 50–100 homoplasy steps in *M. tuberculosis* population, are particularly noisy due to the low number of SNPs in this population arising 50–100 times in the phylogeny (only 9 variants), which makes power estimates of such a small sample to fluctuate.

*M. tuberculosis*, lower sample sizes were required in the single-clade population compared to the species-wide, to detect SNPs of the same MAF, effect size and degree of homoplasy (Supplementary Table 4). In *K. pneumoniae*, we observed the opposite, higher sample sizes were required in the single-clade population. In *E. faecium*, similar sample sizes were needed in both populations to detect highly homoplasic SNPs (acquired 10–50 and 50–100 times) but higher in the single-clade population to detect low-homoplasic SNPs (acquired 1–5 and 5–10 times).

## Discussion

In this work, we showed how existing collections of bacterial genomes can be harnessed to conduct power calculations for bacterial GWAS. Investigators can apply our approach as part of their study design to determine how many strains they would need to sequence and/or phenotype to successfully identify statistically significant associations. Power calculations can also be applied *post hoc*, to report on the limit of detection in terms of the lowest MAF and effect sizes detectable by GWAS in the bacterial population of study. Either way, conducting power calculations will require making a set of assumptions as to the type of causal genotypes (i.e. caused by the acquisition of genes or mutations), their MAF, effect sizes and heritability.

We implemented two approaches to perform power calculations, here labelled as sub-sampling and phenotype simulation approaches. The former requires a known genotype-phenotype relationship in the population of study. We chose antibiotic susceptibility phenotypes, as determined by in vitro susceptibility testing, as they were readily available for the strain collections we used; and because the genetic basis of AMR phenotypes is generally well understood. The advantage of using AMR genes is that

they are common and have high effect sizes; thus, the original population can be sub-sampled and AMR phenotypes modified to achieve the desired reduction in MAF and effect sizes. The limitations are that AMR phenotypes may not always be available or balanced in the population of study. This analysis is also constrained by the underlying genetic architecture, and the maximum MAF and effect size of causal genotypes. To overcome these limitations, we implemented the phenotype simulation approach which allowed us to test the effect of a wider range of parameters (i.e. effect size and heritability) by simulating phenotypes from existing genetic variants of different MAF and homoplasy.

Although it was expected that higher values of these variables (i.e. MAF, effect sizes, heritability and degree of homoplasy) would lead to an increase in power, we were able to determine the exact sample sizes for different combination of parameters and in different populations. Sample sizes required to detect causal acquired genes in a pan-genome GWAS were similar regardless of the bacterial species and population. This was not the case for the detection of causal SNPs, which depended heavily on the population. This may be due to pan-genome sizes being relatively comparable between species and populations, as opposed to a higher degree of genetic diversification in chromosomal genes (number of SNPs). Here we show that MAF, homoplasy and effect size of causal SNPs all have a measurable effect in the sample sizes required to detect them. The fact that the magnitude of such effect varies by population points to population-specific factors having an influence too. Factors that may affect the performance of GWAS, and thus the estimated power, include the accuracy of phylogenetic reconstructions, which may be challenging to obtain in bacteria with high recombination rates, the degree of population stratification and patterns of linkage disequilibrium. These points highlight the fact that the power estimated using samples from one population may not hold true for others, and support our recommendation to conduct power calculations using real genomes for the specific bacterial population of study, which will capture real patterns of population structure and linkage disequilibrium. This should be feasible for well-studied organisms given the large and increasing availability of whole-genome sequenced strains in public repositories.

Our study has several limitations. Here we used Snippy as the bacterial SNP-calling pipeline[34] of choice, as it has been shown to minimise false positives calls[35]. However, a limitation of Snippy is that missing SNP calls are not retained in Snippy's output consensus sequence, so SNP MAF were calculated without considering SNP missing calls. Still, we expect this to have little effect on power estimates, as SNPs chosen to simulate phenotypes are randomly selected multiple times, which means that the power calculated across simulations will account for the multiple and most common missing call rates present in the population. We used a single GWAS tool (PySeer)[24] and method (linear mixed model) to conduct GWAS. Recent work has shown that the power to detect causal variants depends on the GWAS method employed[36]. It was out of the scope of our study to benchmark and give recommendations on individual GWAS methods. Our pipeline can be easily adapted to accommodate other GWAS tools, as multiple steps in the pipeline, such as variant sampling or phenotype simulations, are independent of the GWAS method used. A requirement to run the phenotype-simulation approach is that the ancestral state of genetic variants can be reconstructed so that these can be selected based on their degree of homoplasy. This prerequisite may not be possible for bacterial populations with high recombination rates. Another limitation is that sequenced genomes from the population of interest must be already available. However, for species that are not well-studied, power calculations can be performed by simulating bacterial populations from a reference genome, as previously described[36].

Further work is needed to perform power calculations for complex bacterial phenotypes involving multiple loci (e.g. epistatic effects). It was out of the scope of this work to investigate and report false positives rates. False positives in bacterial GWAS may arise from a variety of reasons, including the degree of genotype missingness, sequencing batch effects, regions in the genome that are hard to genotype, or the degree of stratification and linkage disequilibrium of causal variants in the population. Future work is needed on how best identify and account for factors that give rise to false positives in bacterial GWAS.

In conclusion, power estimates will only apply to the bacterial population for which power calculations were performed from, and may not be generalisable to other populations. We thus recommend the use of existing genomes of the species and population of interest. In our approaches, only phenotypes were changed or simulated – bacterial genotypes and populations were not in any case simulated or modified. Given the inherent differences among bacterial populations, we believe our approach will yield realistic estimates of the sample sizes required to conduct successful GWAS.

## Methods

**Choice of bacterial populations**. Three bacterial species were chosen to represent a gram-positive organism (*E. faecium*), a gram-negative (*K. pneumoniae*) and a species of limited genetic diversity (*M. tuberculosis*), respectively. For each species, a population representative of the species-wide strain diversity was obtained from available collections of whole-genome sequenced strains. A subset of this species-wide population with lower strain diversity, made up of samples from a single clade, was additionally selected (single-clade population).

For *E. faecium*, we used a strain collection isolated from a variety of sources in the UK[37] of 1432 isolate genomes from livestock and meat ($n = 256$), wastewater treatment plants ($n = 383$), bloodstream infections ($n = 782$) and 11 NCTC strains. The *E. faecium* single-clade population consisted of 761 clade A1 isolates drawn from a haematology study[38], where the original population (of 1477 isolates) was de-duplicated to retain a single isolate per subtype (strain type) and patient, and reduced down to 227 and 534 isolates from faecal and ward environmental sources, respectively. To increase the population of clade A1 genomes, we additionally included genomes from a UK hospital study ($n = 292$)[39] and a UK nation-wide ($n = 478$)[40] studies, which increased the population size to 1531 clade A1 genomes.

The *K. pneumoniae* collection was assembled from seven studies[41–47] and consisted of 2628 isolates. The selected publications describe nation-wide or international isolate collections and include human, animal and environmental samples from Africa, Asia, the Caribbean and Europe including their antimicrobial susceptibility phenotypes. We did not include collections from a single source, e.g. a single hospital, to reduce the possibility of including closely related isolates with low genetic diversity originating from a local outbreak. The *K. pneumoniae* single-clade population consisted of 239 CC258 isolates from this collection, plus 396[48] and 558[49] CC258 isolates from an extra two studies.

For *M. tuberculosis*, we chose a collection of clinical isolates ($n = 2655$) generated as part of a global drug resistance project[2] and representing the four main human-infecting *M. tuberculosis* complex (MTBC) lineages (lineages 1 to 4). The *M. tuberculosis* single-clade population consisted of 1139 isolates belonging to the most common sub-lineage (sub-lineage 4.3) in this global collection.

**Genome analysis pipelines**. Raw sequencing data was analyzed with the goal of obtaining three standard files: a multi-sample VCF, a pan-genome table and a phylogenetic tree. For *K. pneumoniae* genomes, Kleborate v1.0.0[50] was used to determine the species of all isolates, of which those with a weak species match or a match to any species other than *K. pneumoniae* were excluded. Genomes with >1000 contigs were also excluded. For *M. tuberculosis*[2] and *E. faecium*[37,38] genomes, the quality control filters applied to discard bad quality genomes are described in the original publications.

For all collections, draft assemblies were generated using an automated *de novo* assembly pipeline based on Velvet[51] and annotated using Prokka[52] v1.11. Pan-genomes were computed using Panaroo[53] v1.2.3 with strict stringency mode. Reads were mapped to the reference genome of each bacterial organism using Snippy v4.6.0 (https://github.com/tseemann/snippy), specifically, the *K. pneumoniae* HS11286 (NC_016845.1), *E. faecium* Aus0004 strain (CP003351), and *M. tuberculosis* H37Rv (GenBank accession NC_000962.3) reference genomes were chosen. Snippy was used as it is recommended as a general purpose bacterial SNP-calling pipeline[34] and has been shown to minimise false positives[35]. Whole-genome alignments were created by keeping a version of the reference genome with only substitution variants (i.e. SNPs but not indels) instantiated (i.e. Snippy's .consensus.subs.fa output file). Single-nucleotide polymorphisms (SNPs) were extracted from whole-genome alignments using snp-sites[54] v2.5.1 and saved as a multi-sample VCF files (the one used for power

calculations). Missing SNP calls were not retained as these are not specified in Snippy's .consensus.subs.fa files used. IQ-TREE v1.6.10 was used to create a maximum likelihood tree from the core-genome alignment. Recombination events (including potential mobile genetic elements) were detected by Gubbins v1.4.10[55], using an appropriate outgroup and the IQ-TREE phylogenetic tree as the starting tree. The final tree produced by Gubbins, obtained by running RAxML v8.2.8[56] without recombination regions, was rooted on the outgroup and kept for further analyses (i.e. ancestral state reconstruction and population structure calculations). Linkage disequilibrium was measured with the r-squared metrics calculated by Plink[57] for each population.

### Selection of known genotype–phenotype relationships.
Known phenotype–genotype relationships can be used to establish if a GWAS is able to identify known causal variants within a given bacterial population. We chose antibiotic susceptibility as the bacterial trait of study because (1) antibiotic susceptibility phenotypes, as determined by in vitro susceptibility testing, are readily available for many strain collections; and (2) the genetic causes of antibiotic susceptibility are generally well understood. We chose antibiotics that met the following criteria: (1) had a balanced proportion of resistant and susceptible isolates; (2) were caused by either the acquisition of genes or mutations in chromosomal genes; and (3) there was a good correlation between the presence of a well-known antibiotic resistance-conferring gene (or mutations) and antibiotic susceptibility in the studied collections, as determined by a large odds ratio (Table 1). Acquired AMR causal genes were identified running Pyseer v1.3.5[24] with Roary's output pan-genome file as genotypes and AMR susceptibilities as phenotype, and BLASTing the top hits against the CARD database[58]. For *M. tuberculosis*, we identified causal AMR mutations as those in the VCF file also found in the TBProfiler database[32,33]. The MAF and effect size of AMR causal variants were calculated (Table 1) and used as the maximum values in the sub-sampling approach.

### Implementation of PowerBacGWAS pipeline: data pre-processing steps.
Three input files are needed to run the power calculations pipeline: a multi-sample VCF file, a Roary-formatted pan-genome CSV file and a phylogenetic tree in Newick format (Supplementary Fig. 4). The format of the VCF file can be checked using the script prepare_vcf_file.py which will make sure multi-allelic sites are split, variant identifiers (ID VCF field) are included and genotypes (GT VCF field) are converted to haploid format. The internal nodes of the phylogenetic tree need to be annotated, i.e. labelled with unique identifiers, using the script annotate_nodes_newick.py. The Roary-formatted file must correspond to the gene_presence_absence.Rtab file (http://sanger-pathogens.github.io/Roary/), a simple tab delimited binary matrix with the presence and absence of each gene in each sample. Because of the use of PastML[59] and PLINK[57] tools in the pipeline, VCF and Roary-formatted files need be converted to a CSV file compatible with PastML (https://pastml.pasteur.fr/help) using vcf_to_pastml_matrix.py and roary_to_pastml_matrix.py scripts, respectively; and to PLINK binary PED files (https://www.cog-genomics.org/plink/2.0/input#bed) using the scripts vcf_to_plink_files.py and roary_to_plink_files.py (Supplementary Fig. 4). PLINK input-formatted files are needed to run GCTA[60] phenotype simulations.

### Implementation of PowerBacGWAS pipeline: ancestral state reconstruction.
Because one of the goals of the pipeline is to assess the power of detecting causal variants depending on their degree of homoplasy, the ancestral state of variants need to be reconstructed to count how many times they arose in the phylogeny. We used PastML[59] v1.9.20 to infer the ancestral characters of VCF variants and genes using maximum parsimony. Because ancestral state reconstruction is computationally expensive but, at the same time, it is applied independently to each variant, we sliced input variant files to parallelise this process using multi-processing. The wrapper scripts ancestral_state_reconstruction.py and ancestral_state_reconstruction.roary.py implement this step for VCF and Roary files, respectively (Supplementary Fig. 4). We additionally calculated the number of independent variant acquisitions (homoplasies) at the region (e.g. gene) level using the script calculate_changes_per_region.py.

### Implementation of PowerBacGWAS pipeline: sub-sampling approach.
The sub-sampling approach uses a known genotype-phenotype relationship to conduct power calculations and is implemented for binary phenotypes. It supports power calculations for acquired genes (pan-genome GWAS), VCF variants (variant GWAS) and burden testing, where VCF variants are aggregated within chromosomal regions (e.g. genes), and regions tested for association. The scripts prepare_gwas_runs_subsampling.py and prepare_gwas_runs_subsampling_roary.py implement this approach for VCF variants and the pan-genome, respectively. In addition to the variant file (VCF or pan-genome), this script requires a file with causal variants (a GCTA-formatted file with a list of variants IDs) and a Pyseer-formatted phenotype file, where the variants in the causal variants file are known to determine the trait in the phenotype file. A parameters file is also required to specify the range of MAF, sample sizes and effect sizes to test. First, the observed MAF and effect size of causal variants are calculated. Second, and for each parameter combination, the original phenotype file is modified to 'artificially' decrease

the MAF (the proportion of samples carrying causal minor allele) and the sample size (total number of samples considered) by assigning 'NA' phenotype labels to as many isolates as needed to achieve the chosen reduction in sample size and MAF. The effect size is reduced by swapping phenotype labels so that the number of false positives and false negatives increases at the expense of a reduction in the number of true positives and true negatives; only between isolates of the subsample to ensure MAF and sample size remain unchanged. This sub-sampling step is repeated as many times for each parameter combination as specified in the parameters file. Finally, prepare_gwas_runs_subsampling.py outputs a bash script with GWAS runs (as many as sub-samples) and a CSV file with all parameter combinations used. Pyseer v1.3.5[24] was used to run the GWAS using the linear mixed model (LMM) to account for population structure. The kinship matrix for LMM is calculated from the phylogenic tree using PySeer's script phylogeny_distance.py and thus is kept the same for all GWAS analyses (SNP, burden testing and pan-genome GWAS) applied to the same population. Finally, the script process_gwas_runs.py is used to extract the LMM adjusted *p*-values of causal variants from Pyseer output files; and the R script plot_gwas_runs_subsampling.R to plot the results.

### Implementation of PowerBacGWAS pipeline: phenotype simulation approach.
We implemented a second approach to conduct power calculations for when a genotype–phenotype relationship is not known in the population, for both binary and quantitative phenotypes. In brief, this approach consists in sampling existing genes from the pan-genome, variants from a VCF file or mutated regions (i.e. for burden testing) meeting a predefined MAF and degree of homoplasy, to then simulate phenotypes from these variants with a pre-defined effect size and sample size. The wrapper scripts prepare_gwas_runs.py and prepare_gwas_runs_roary.py implement this approach for VCF variants and the pan-genome, respectively (Supplementary Fig. 4). In addition to the variant file (VCF or pan-genome), this script requires a table with the number of homoplasies per variant, as produced by scripts ancestral_state_reconstruction.py or ancestral_state_reconstruction.roary.py. A parameters file is also required to specify 'causal variant sampling parameters' (that is, range of MAF to test, level of homoplasy, number of causal variants and sampling repetitions) and 'phenotype simulation parameters' (range of sample sizes and effect sizes to test, and simulation repetitions, among others). The units of effect sizes are specified as odds ratios when simulating binary phenotypes or in beta units when simulating quantitative phenotypes. For a given combination of parameters, the script sample_casual_variants_from_vcf.py, called within the wrapper script prepare_gwas_runs.py, reads a VCF file and randomly samples the chosen number of causal variants that meet the indicated MAF and homoplasy level. The script outputs the list of variant IDs along with their chosen effect size in a GCTA-compliant format. The number of times this variant sampling step is repeated can be chosen in the parameters file. This was set to 10 per set of simulation parameters in the analyses presented in this manuscript. If the option of a burden test is selected, then the script outputs all variants in the regions meeting the chosen MAF (proportion of samples with any variant within a region) and homoplasy criteria (number of independent homoplasies within a region, computed by script calculate_changes_per_region.py). The script simulate_phenotype_using_gcta.py reads the list of causal variants previously sampled, and uses the additive genetic model implemented in GCTA[60] to simulate phenotypes. When simulating binary phenotypes with GCTA, we noted that the variable that had the biggest impact on the distribution of mutated and wildtype individuals among cases and controls, and thus association *p*-values, was the heritability, not so much the effect size (odds ratio) specified in the causal variants file. We thus decided not to use GCTA when specifying odds ratios to simulate binary phenotypes, but only when specifying heritability values or simulating quantitative phenotypes. The script simulate_binary_phenotype_vcf.py (and simulate_binary_phenotype_roary.py) was written to implement a custom method to simulate binary phenotypes for a given odds ratio.

This script takes the multi-sample VCF and list of causal variants as input files; and the chosen number of cases and controls (sample size), MAF and odds ratio to be simulated as parameters. The function scipy.optimize.least_squares is used in this script to solve the following set of equations:

$$E1 : \text{sample\_size} * \text{MAF} = \text{mutated\_controls} + \text{mutated\_cases}$$

$$E2 : \text{sample\_size} * (1 - \text{MAF}) = \text{wildtype\_controls} + \text{wildtype\_cases}$$

$$E3 : \text{cases} = \text{mutated\_cases} + \text{wildtype\_cases}$$

$$E4 : \text{controls} = \text{mutated\_controls} + \text{wildtype\_controls}$$

$$E5 : \text{odds\_ratio} = (\text{mutated\_cases}/\text{mutated\_controls})/(\text{wildtype\_cases}/\text{wildtype\_controls})$$

Where variables sample_size, MAF, cases (number of cases), controls (number of controls) and odds_ratio are known (i.e. specified by the user); and variables, mutated_cases, wildtype_cases, mutated_controls and wildtype_controls, defining the number of cases and controls with and without causal variants, are calculated. The script identifies which samples (i.e. isolates) in the VCF file carry the causal alleles (mutated) and which ones do not (wildtype). Then, it randomly selects the number of mutated_control and mutated_cases from the pool of mutated samples and labels them as controls and cases, respectively; and selects the number of

wildtype_controls and wildtype_cases from the pool of wildtype samples and labels them as controls and cases, respectively. This is how the phenotype labels are simulated to achieve the specific sample size (with a particular ratio of cases and controls), MAF and odds ratio chosen by the user.

The number of times this phenotype simulation step is repeated can be chosen in the parameters file. This was set to 10 per set of simulation parameters in the analyses presented in this manuscript. The output of the phenotype simulation step is a Pyseer-compliant phenotype file for a given parameter combination. Finally, the wrapper script prepare_gwas_runs.py outputs a bash script with as many GWAS runs, and a CSV file with as many lines, as parameter combinations. As done for the sub-sampling approach, Pyseer v1.3.5[24] is used to run the GWAS using the linear mixed model (LMM) and the script process_gwas_runs.py to extract the LMM adjusted $p$-values of causal variants from Pyseer output files. The R script plot_gwas_runs.R is used to plot the strength of association of causal variants in the y-axis (i.e. $-\log_{10}(p\text{-value})$, as plotted in Manhattan plots) as a function of sample sizes (along the x-axis) for different combinations of effect sizes and MAF. The results can also be plotted for different combination of heritability values and MAF. Note that the pipeline (in the parameters files) allows to vary the effect sizes or heritability values of causal variants, but no both at the same time. The effect of varying degrees of homoplasy on the association strength can also be plotted across sample sizes for a fixed MAF. For each set of parameters, the total number of GWAS replicates is determined by the number of times the variant sampling step is repeated multiplied by the number of times the phenotype simulation step is repeated. Power is calculated for each combination of parameters as the proportion of GWAS replicates in which the causal variant is above the Bonferroni-corrected genome-wide significance threshold.

We used a single causal variant (i.e. single acquired gene in the pan-genome GWAS; single SNP in the SNP GWAS and single mutated gene in the burden GWAS) to simulate phenotypes, to assess the limit of detection of individual causal variants. The pipeline does not currently support the simulation of phenotypes from multiple causal variants.

**Docker and Nextflow implementation of PowerBacGWAS**. Given the multiple software tools, modules and package dependencies of PowerBacGWAS, we built a Docker image of the pipeline (https://hub.docker.com/r/francesccoll/powerbacgwas) to facilitate usage. We have additionally implemented a Nextflow pipeline that uses this Docker image, to automate the multiple computational steps and parallelise the GWAS runs. Using the Nextflow implementation, the pipeline is reduced to only three steps: (1) preparation of input files, (2) ancestral state reconstruction and (3) GWAS runs. In the latter, the user can choose the type of variants, phenotype, GWAS method and approach to power calculations. The GitHub page (https://github.com/francesccoll/powerbacgwas)[61] includes instructions on how to install PowerBacGWAS via Docker and Nextflow, or locally. The Usage Wikipage (https://github.com/francesccoll/powerbacgwas/wiki#usage) includes sections on how to run the pipeline using the Docker/Nextflow installation ('Nextflow commands') or how to run individual scripts using the locally installation ('Individual commands'). The computational time of running PowerBacGWAS depends on the number of parameter combinations, and number of variant sampling and phenotype simulation repetitions indicated in the parameters file. All these parameters combined determine the number of individual GWAS runs. We recommend to run the Nextflow pipeline with the LSF executor, if an LSF cluster is available, for faster running times, wherein each process is submitted as a separate job. Overall, for the six bacterial datasets used in this work, PowerBacGWAS used a median of 1,670 CPU hours (interquartile range: 548 to 3542) and a median duration of 3.1 h (interquartile range: 2–7.6 h) when using the Nextflow LSF executor.

**Reporting summary**. Further information on research design is available in the Nature Research Reporting Summary linked to this article.

## Data availability

The whole-genome sequences of the strain collections used in this study are available on European Nucleotide Archive (ENA) under the accessions listed in Supplementary Data 1, which also includes the antibiotic susceptibility phenotypes. Processed data (VCF, pan-genome, phylogenetic trees, phenotype, causal variants and parameters files) used to generate the PowerBacGWAS results presented in this manuscript are available on GitHub (https://github.com/francesccoll/powerbacgwas/tree/main/data). The source data of Figs. 2 and 3 can be found in Supplementary Data 2 and 3, respectively.

## Code availability

All scripts necessary to run the power calculations pipeline are available on GitHub (https://github.com/francesccoll/powerbacgwas)[61]. Docker images are available on https://hub.docker.com/r/francesccoll/powerbacgwas.

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

## Acknowledgements

This project was funded by Wellcome Trust grant (201344/Z/16/Z) awarded to Francesc Coll. This publication was supported by the Health Innovation Challenge Fund (WT098600, HICF-T5-342), a parallel funding partnership between the Department of Health and Wellcome Trust. T.G.C. is funded by the Medical Research Council UK (Grant no. MR/M01360X/1, MR/N010469/1, MR/R025576/1 and MR/R020973/1) and BBSRC UK (Grant no. BB/R013063/1). The views expressed in this publication are those of the author(s) and not necessarily those of the funders.

## Author contributions

F.C. designed the study with input from J.Pa.; F.C. undertook the bioinformatic analyses with contributions from T.G., S.B. and J.Ph.; T.G. and K.E.R. curated, analyzed and provided the *E. faecium* collections used in this study whereas S.B. did so for the *K. pneumoniae* collections. J.Ph. and T.G.C. generated, analyzed and provided the *M. tuberculosis* collections. F.C., J.Pa. and S.J.P wrote the first draft of the manuscript. S.J.P. and J.Pa. supervised the study. All authors had access to the data and read, contributed and approved the final manuscript.

## Competing interests

The authors declare no competing interests.
