## [Peer Review File · Communications Biology]

Reviewers' comments:

Reviewer #1 (Remarks to the Author):

The manuscript from Coll et al, describes two novel approach in measuring the power of bacterial GWA studies. They described their results in an informative paper along with set of python, perl and R scripts implementing their two approaches. They also developed a novel method to simulate binary phenotype in bacterial population. Using the developed computational pipeline (powerBacGWAS,) they evaluated the effect of minor allele frequency, effect size and homoplasy on power of pyseer LMM-based GWAS to detect causal markers in different sample sizes. This work was much needed and can be an essential part of future bacterial GWA works to check whether the collected samples are sufficient to identify causal markers of certain effect sizes and can also be used to roughly estimate required sample size in designing bacterial GWA studies. While the authors have correctly evaluated the effect of important determinants of power in bacterial GWAS such as allele frequency, effect size and homoplasy, they need to add some discussion to prevent misinterpretation of the results. Here are some points in the text which need extra explanations for a more clear transfer of findings to readers.

- Line 295-333: It is interesting that authors have performed almost every step of genome analysis differently for the three investigated strains. Since quality of genome analysis and phylogenetic tree construction are important factors in GWA study when using pyseer, variation in these steps might lead to variation in estimated power. Therefore, it will be informative to also discuss the role of the used genome assembly, variant calling and phylogenetic tree constructions methods in power of each bacterial GWAS and pointing out the fact that, in order to use the power estimates presented in this paper, users need to use the same methodology for their dataset.
- Authors of pyseer in their recent work have shown the better performance of elastic-net over mixed models in pyseer, meaning that the estimated power for different bacterial species shown by the authors might be an underestimation. Although that's beyond the scope of this paper to compare different GWAS methods, it will be useful to readers to add explanation why authors have chosen LMM over elastic-net.
- While phylogeny-based relationship matrix is known to have a better performance over genotype-based estimation in pyseer, its performance relies on having accurate phylogenetic tree that is challenging to obtain in bacteria with high recombination rates and could consequently affect pyseer LMM GWAS results and estimated powers. This fact needs to be added to discussion.
- Population stratification is a critical confounding factor in bacterial GWAS as also acknowledged by authors and while pyseer-LMM is successful to adjust for this factor, to some extent, the performance of pyseer LMM GWAS varies based on the level of stratification in the population under study. In other words, the estimated power using samples from public databases may not hold true for other bacterial populations in case they have different levels of stratification. This point should also be clarified in the discussion.
- Line 153 and 182: The power estimates for single-clades for single-clade *M. tuberculosis* seems noisy (Figure 3e)! This might be due to strong genome-wide linkage disequilibrium or stratification of this dataset. Specifically, strong genome-wide linkage-disequilibrium (LD) is a well-known characteristic of *M. tuberculosis* genome and may partially explain the different results obtain for this species and need to be discussed in the discussion. In general, LD would have been a better parameter than diversity to compare between the investigated species because it is known to be an important confounding factor in microbial GWAS.
- Line 398: Developing a novel approach for binary phenotype simulation by modifying a set of known phenotypes(simulate_binary_phenotype_vcf.py) is one of the interesting novelties of the paper, however, it is not fully explained how it is done. It would be informative to readers if it is explained (using mathematical equation or pseudocode) how minor allele frequency of a known causal variant is 'artificially' decreased to reach desired value without modifying the genotype data along with the mathematical equation based on which effect size of a causal marker is reduced by swapping phenotype labels.
- Line 434-437: This is a very interesting observation! Considering that GCTA is one of the most commonly used tools for phenotype simulation it would be helpful to the community to make a direct comparison between the phenotype simulation equation used by GCTA (<https://cnsgenomics.com/software/gcta/#GWASSimulation>) and equation user here and discuss the factors that contribute to the better performance of the model developed here.
- It is not clear whether the results of phenotype-simulation based power estimation are based on

quantitative or binary phenotype simulation. Should be added to methods and results section.
Minor points:

- Line 33: Meaning unclear! Does it mean modifying the known relationship to fit different scenarios?
- Line 108: powerBacGWAS does not predict phenotype labels so should it mean "recover the simulated causal markers"
- Lines 229-230: meaning unclear! Should be rephrased.
- Lines 257-258: meaning unclear!
- Line 333: "Roary's pan-genome analysis was not performed for the M. tuberculosis collection". Explanation needed why this analysis and consequently pan-genome GWAS was not performed for M. Tuberculosis.
- Figure 1: figure legend not readable!
- Figures 2 and 3: These figures will be more informative if the variations across 10 replications of data points are presented by error bars.

Technical points:

powerBacGWAS tool is the main finding of the paper and can significantly improve the quality of future bacterial GWAS. The codes of powerBacGWAS are well-written and well-annotated, however, their implementation as a tool need two main improvements:

1) Workflow management: powerbacgwas produces lots of intermediate files which makes it difficult to use just by running commands one after another manually. Workflow management tools are developed to solve this problem and make the tool much more user-friendly by eliminating the need to manually handle the intermediates.

2) Parallelization: Although, powerBacGWAS parallelizes ancestral state construction, it does not parallelize the time-consuming process of all the GWAS runs which would make it difficult to use for power estimation in scenarios with lots of markers and large sample size.

I would recommend implementing powerBacGWAS as a workflow using management tools such as snakemake which can solve both handling of intermediates and parallelization of all the steps. I believe that developing a user-friendly version of the tool would significantly improve the impact of this paper as well. Nextflow, Airflow and other workflow management tools may also be good candidates.

Minor points:

1. Following two dependencies are used by the tool but not mentioned in GitHub page. Need to be added to dependencies.
 - a. Perl
 - b. Bioperl (use for internal node annotation)
2. "Three input files are needed to run the power calculations pipeline": Pan-genome GWAS and SNP-based GWAS are independent of each other (as shown in supplementary fig. 1) and users may need to just perform one. Explanation needed why both multi-sample VCF file and Roary-formatted pan-genome CSV file are needed to run powerBacGWAS.
3. In tutorial (<https://github.com/francescoll/powerbacgwas/wiki#usage>), the subsampling approach, the command for running SNP-based approach is missing!
4. In parameter file:
 - a. Mentioned: "NOTE: aphA gene observed causal variant odds ratio 89.10721003134796, thus 89 chosen as maximum" but maximum effect size is set to 100! Clarification needed!
5. In parameter file used by 'prepare_gwas_runs.py' which includes phenotype-simulation based power estimation using GCTA, it is not clear how to differentiate between binary and quantitative phenotype simulation. In other words, how powerBacGWASim is set to calculate binary or quantitative power estimates?
6. In simulate_binary_phenotype_vcf.py line 377:
 - a. 'roary_samples_mut' and 'roary_samples_wt' are not defined before using! (probably a bug due to copying script lines from simulate_binary_phenotype_roary.py which leads the code to fail)

Reviewer #2 (Remarks to the Author):

In their manuscript entitled "Alternative approaches to conduct power calculations for bacterial genome-wide association studies" Coll et al. present a new tool, PowerBacGWAS, which allows users to perform simulations using existing genomes to determine the effects of a range of input parameters on the ability to detect causal variants, and the effect sizes detectable with given sample sizes. The authors have implemented two simulation approaches, one which subsamples a dataset with a known phenotype, and the other simulates the phenotypes for a set of genomes. The utility of the tool is demonstrated by performing a large number of simulations testing the effect of a range of parameters including sample size, allele frequency, effect size, heritability and homoplasy in collections of three bacterial species. Using collections of existing genomes means that simulations are performed using genomes with real patterns of important parameters such as population structure and linkage disequilibrium. The pipeline is available on GitHub with a detailed tutorial.

PowerBacGWAS has the potential to be a useful tool to the bacterial GWAS community. However to improve its interpretability I think the tool should either add to or change the value reported from the simulations. The manuscript describes the tool as performing power calculations. Currently the mean p value per set of simulations is reported and presented in the manuscript, however this is not power, which is the ability to correctly reject the null hypothesis, i.e. the proportion of the simulations in which the true/simulated causal hit is significant. More detail around which parameters changed and remained constant across the simulations would also improve understanding and comparison between the results.

Detailed comments

1. The value reported is the average (I assume mean) p value for a set of simulations, but this doesn't tell you the power, the probability that the null hypothesis will be correctly rejected for a set of parameters. A more informative way to report the results would be to report the % of simulations where the causal variant is significant per set of parameters. Or for the tables, the number of genomes required for a set of parameters to give a specified level of power.
 - a. As the results are currently reported, some sets of parameters are reported as 'non-detectable using GWAS', however the mean p value of the causal variant being below the significance threshold does not necessarily mean that power was zero.
 - b. Likewise where it is reported that a given sample size is enough to detect a variant at a particular effect size, this is true in that the results mean that power is non-zero, but it does not tell you power itself. Users may want to know the increase in power for an increase in sample size above the level at which an effect is detectable. Power is more informative on study design than the mean p value attained by simulations.
2. It would also be very useful if the false positive rate could also be reported for the simulations alongside power. Or a statement made on whether a low FPR is maintained as power increases with a given parameter.
3. The authors describe the results of a range of different parameters that impact the power to detect causal variants. However when a parameter is changed, it needs to be clearer what the values are for the other parameters, and if they are kept constant while the parameter of interest is being changed. In particular, when the effect size is being assessed, is heritability kept constant and at what value? When heritability is being assessed, is the causal variant effect size kept constant and at what value?
4. I think the language around the impact of variant frequency needs to be clearer, it is not the frequency of a particular variant but the minor allele frequency of either presence/absence of the variant that determines power. The language in the manuscript seems to suggest that the higher the frequency the higher the power E.g. line 144.
 - a. On this note, Supplementary Table 1 legend states that the complete *E. faecium* single-clade collection was not used because the resistance conferring gene was 79% frequent, too high to perform power calculations. It is not clear why this is too high.
5. Burden testing
 - a. How were the burden tests performed?
 - b. Line 179: 'could detect variants of down to 2.5% frequency' what does variant mean in this

case? Is this discussing a joint burden of mutations that together add up to 2.5% of genomes affected, or does this mean it can detect a gene containing multiple variants, the lowest of which is individually found at 2.5% frequency?

c. Line 177-178: It is not the case for all parameter combinations in Table 3 (where effect size is adjusted) that the number of genomes required is smaller for the burden GWAS.

d. Supplementary Table 3 (where heritability is adjusted) the burden GWAS is sometimes able to detect effects at lower frequencies than the variant GWAS is able to, but where both are able to detect an effect at a particular frequency, the burden GWAS often requires a much larger number of genomes which feels contradictory, could this be explained? Is the simulated effect size being kept constant for these simulations?

6. Line 180-181: A larger number of genomes are required to detect the same odds ratio for SNPs versus genes, is this due to the higher multiple testing correction threshold or do the authors think there is another reason for this?

7. What is the kinship matrix for the LMM calculated from? Is the kinship matrix kept the same for the SNP versus gene analysis per species?

8. Line 442-444: More detail is required on the method to simulate binary phenotypes and the differences to the GCTA phenotype simulations explained. Why does heritability not have such a strong effect when using this method? Is heritability kept constant through these simulations, if so what value is it set to?

9. Figure 3: Could the authors explain why the mean p values for TB at 50/100 homoplasmy steps oscillate so much in significance with changing sample size compared to the other simulations?

10. It appears that multiple causal variants can be simulated. In the case where more than one causal variant is simulated, are these modelled additively? When the variants are tested individually (not by a burden test) how does the pipeline report the result of the simulation?

11. Samples and SNP sites with an excess of 10% missing calls are removed, but how are the remaining missing calls handled in the GWAS? If missing calls are ignored when a site is tested then the simulations will be comparing different numbers of genomes depending on the call rate of the causal variants selected.

Minor comments

12. Line 108: Should this say 'identify the sample size needed to recover the simulated causal variant' instead of 'identify the sample size needed to recover the simulated phenotype'?

13. Line 130, 131 and 133: Are these the allele frequencies in the whole population or in just those phenotyped for the antibiotic in question, i.e. is this the maximum frequency that could be tested in the phenotyped dataset or was that lower?

14. Line 485-486: How were the number of independent tests defined for the Bonferroni correction?

15. Line 176-177: There is a supplementary figure for the burden test results but not for the SNP GWAS results, could a figure be added for the SNP results?

16. Line 213: It will also require knowing or making an assumption on the heritability of the phenotype

17. The methods need to state number of repeats per set of simulation parameters

Reviewer #3 (Remarks to the Author):

In this paper, Coll and colleagues introduces PowerBacGWAS, a bioinformatic approach to estimate

statistical power in bacterial GWAS. This is an interesting topic and it is easy to see the potential need for such a tool in the field. However, I have some concerns about the statistical rigor of this study which I will illustrate below.

Major comments:

1. The authors seem to have misunderstood the concept of statistical power, which is unfortunate for a paper about power calculation. Given a type-I error rate (e.g., 0.05, or a Bonferroni-corrected threshold), statistical power is the probability for a test to give a p-value below this cutoff under the alternative hypothesis. Therefore, if we fix the level of statistical power we want (e.g., 80%), needed sample size can be calculated. Similarly, if we fix the sample size (along with effect size and other factors), power can be estimated as a percentage between 0 and 1. This is not how this study approached this problem. Using the proposed "subsampling approach" as an example, the authors basically started from a known genotype-phenotype association with large N, then reduced N (let's assume the effect size to be fixed for simplicity), and calculated one p-value for each corresponding N. If the p-value reaches the significance threshold, the authors claim to have had enough power. This approach completely ignored the fact that p-value itself is a random variable and if the study is repeated in an independent dataset with all the identical parameters, p will be different and may or may not be significant. Although in some simple scenarios, this issue may be resolved by simply repeating the subsampling procedure many times to get an empirical distribution of p-values, it gets trickier if the subsampled N is relatively large since different repeats will have substantial sample overlap and therefore the p-values obtained from these repeats won't be independent.

2. Throughout the Methods section, the authors focused on describing the bioinformatic pipeline, mentioning numerous pieces of scripts. However, no statistical models were presented. What is the linear mixed model used to conduct GWAS in this study? What are the fixed/random effects in the LMM? If the GWAS approach is just a LMM, why can't general power calculation tools designed for LMM be used to estimate power for bacterial GWAS? How is bacterial GWAS special compared to any general LMM application? The authors claimed that power calculators for human GWAS cannot be applied due to unique characteristics of bacterial populations (clonal reproduction, strong population structure, and uneven and varying degrees of recombination), but it is unclear why this is the case without showing the model.

3. GCTA was used to simulate phenotypes in the proposed "phenotype simulation approach". GCTA was designed to model highly polygenic complex traits with random genetic effects equally attributed to all variants in the model. Is this compatible with the genetic architecture of bacteria phenotypes? Although technically it is possible to specify the causal SNP effect sizes and simply use GCTA to obtain the error term, some details are still lacking such as how many causal SNPs are there and if they all have the same pre-specified effect sizes?

4. The section about simulating binary phenotypes is a bit odd. Prevalence of binary traits was not mentioned in the paper. Related to this, when heritability was specified in the analysis, is it on the liability scale or the observed scale? It also isn't clear why variants' effect size (odds ratio) didn't matter any more once the heritability was specified. The way it was described in the Methods section was more like an observation than an explanation. In the same paragraph on page 18, it was stated that "this method solves a set of equations to find the number of cases and controls..." and the equations were not shown/explained. Overall, the Methods section (which is what I focused on the most since this is a methodology paper) reads more like a user manual that explains which scripts the users should apply instead of a manuscript's Methods section that explains the statistical and computational details. This needs to be improved.

5. It is unclear why the allele frequencies were specified differently between panels A and B of Figure 2. Also, is it reasonable to specify similar parameter values (for both allele frequency and effect size) in panel C which is about burden tests? By the way, panel C is mislabeled as "A" in the figure.

6. Acronym "AMR" isn't defined in the paper.

7. Causal language was used when explaining GWAS effect sizes (e.g., page 4, line 85). Is bacterial GWAS not aiming to identify indirectly associated markers in LD with the (potentially unobserved) causal variants?

Reviewers' comments:

Reviewer #1 (Remarks to the Author):

The manuscript from Coll et al, describes two novel approach in measuring the power of bacterial GWA studies. They described their results in an informative paper along with set of python, perl and R scripts implementing their two approaches. They also developed a novel method to simulate binary phenotype in bacterial population. Using the developed computational pipeline (powerBacGWAS,) they evaluated the effect of minor allele frequency, effect size and homoplasia on power of pyseer LMM-based GWAS to detect causal markers in different sample sizes. This work was much needed and can be an essential part of future bacterial GWA works to check whether the collected samples are sufficient to identify causal markers of certain effect sizes and can also be used to roughly estimate required sample size in designing bacterial GWA studies. While the authors have correctly evaluated the effect of important determinants of power in bacterial GWAS such as allele frequency, effect size and homoplasia, they need to add some discussion to prevent misinterpretation of the results. Here are some points in the text which need extra explanations for a more clear transfer of findings to readers.

- Line 295-333: It is interesting that authors have performed almost every step of genome analysis differently for the three investigated strains. Since quality of genome analysis and phylogenetic tree construction are important factors in GWA study when using pyseer, variation in these steps might lead to variation in estimated power. Therefore, it will be informative to also discuss the role of the used genome assembly, variant calling and phylogenetic tree constructions methods in power of each bacterial GWAS and pointing out the fact that, in order to use the power estimates presented in this paper, users need to use the same methodology for their dataset.

We had originally obtained the pan-genome, SNP alignments and phylogenetic trees from the co-authors who generated these datasets, hence why different genome analyses had been used. For consistency, and to avoid the variation in power estimates that may arise from employing different genome analysis pipelines, we have now applied the same genome analyses and phylogenetic tree reconstruction to all collections. The Methods section ‘Genome analysis pipelines’ has been rewritten to reflect these changes. Table 1 (column ‘# of SNP sites’ and ‘# of genes in pan-genome’) has been updated accordingly. We have uploaded the latest input files to <https://github.com/francescoll/powerbacgwas/tree/main/data>

- Authors of pyseer in their recent work have shown the better performance of elastic-net over mixed models in pyseer, meaning that the estimated power for different bacterial species shown by the authors might be an underestimation. Although that’s beyond the scope of this paper to compare different GWAS methods, it will be useful to readers to add explanation why authors have chosen LMM over elastic-net.

We had originally used LMM as, at the time the project started, this was the method of choice for bacterial GWAS. We now allow the user of PowerBacGWAS to choose the GWAS method, between LMM and elastic-net, when running power calculations.

- While phylogeny-based relationship matrix is known to have a better performance over genotype-based estimation in pyseer, its performance relies on having accurate phylogenetic

tree that is challenging to obtain in bacteria with high recombination rates and could consequently affect pyseer LMM GWAS results and estimated powers. This fact needs to be added to discussion.

We have now run Gubbins on all six populations to detect and remove recombination prior to phylogenetic reconstruction. This should, in part, remove the effect of recombination on distorting the phylogeny reconstructions. We have added this point to the Discussion (lines 276-280).

- Population stratification is a critical confounding factor in bacterial GWAS as also acknowledged by authors and while pyseer-LMM is successful to adjust for this factor, to some extent, the performance of pyseer LMM GWAS varies based on the level of stratification in the population under study. In other words, the estimated power using samples from public databases may not hold true for other bacterial populations in case they have different levels of stratification. This point should also be clarified in the discussion.

We have extended the Discussion paragraph describing the variables affecting power to discuss this factor (population stratification) too (lines 276-280). Altogether, we think this point reinforces our recommendation to: “conduct power calculations for the specific bacterial species and population of study, as the power estimated using samples from one population may not hold true for others.”

- Line 153 and 182: The power estimates for single-clades for single-clade *M. tuberculosis* seems noisy (Figure 3e)! This might be due to strong genome-wide linkage disequilibrium or stratification of this dataset. Specifically, strong genome-wide linkage-disequilibrium (LD) is a well-known characteristic of *M. tuberculosis* genome and may partially explain the different results obtain for this species and need to be discussed in the discussion. In general, LD would have been a better parameter than diversity to compare between the investigated species because it is known to be an important confounding factor in microbial GWAS.

We have extended Table 1 (which summarises several diversity parameters) to include the mean and interquartile range of the r-squared, a commonly used measure of linkage disequilibrium. We used Plink to calculate pairwise r-squared values between SNPs in the bacterial chromosome.

Figure 3E is noisy specifically for SNPs with 50-100 homoplasy steps at 10% frequency. This is because of the low number of SNPs in this population (*M. tuberculosis* species-wide population) arising 50 to 100 times in the phylogeny (only 9 variants), which makes average p-values (and power) of such a small sample to fluctuate a lot. Many more variants could be randomly sampled with 10-50 homoplasy steps (n=70), 5-10 (n=381) and 1-5 steps (n=468), making the p-value averages (and power) of such larger samples more stable.

Please note that, as recommended by other reviewers, PowerBacGWAS now calculates and reports power as the proportion of simulations in which the true/simulated causal hit is significant for a specific combination of parameters, in addition to average p-values. Figures now plot power, and tables the sample sizes require to achieve 80% power.

- Line 398: Developing a novel approach for binary phenotype simulation by modifying a set

of known phenotypes(simulate_binary_phenotype_vcf.py) is one of the interesting novelties of the paper, however, it is not fully explained how it is done. It would be informative to readers if it is explained (using mathematical equation or pseudocode) how minor allele frequency of a known causal variant is ‘artificially’ decreased to reach desired value without modifying the genotype data along with the mathematical equation based on which effect size of a causal marker is reduced by swapping phenotype labels.

We have added more text explaining our custom approach and the functionality of this script to simulate binary phenotypes, and the set of mathematical equations that are solved in this script. See lines 554-576.

- Line 434-437: This is a very interesting observation! Considering that GCTA is one of the most commonly used tools for phenotype simulation it would be helpful to the community to make a direct comparison between the phenotype simulation equation used by GCTA (<https://cnsgenomics.com/software/gcta/#GWASSimulation>) and equation user here and discuss the factors that contribute to the better performance of the model developed here.

GCTA was designed to model highly polygenic complex traits with random genetic effects equally attributed to all variants in the model. However, we used a single causal variant to simulate phenotypes. This is not to say that bacterial phenotypes are caused by single variants (although this is often the case for antibiotic resistance phenotypes) but to assess the limit of detection of individual causal variants, which is often the unit of interpretation. This may explain why changing the effect size of a single causal variant had no effect on the binary phenotypes simulated by GCTA.

We have now been more explicit in the Methods section: “We used a single causal variant (i.e. single acquired gene in the pan-genome GWAS; single SNP in the SNP GWAS and single mutated gene in the burden GWAS) to simulate phenotypes, to assess the limit of detection of individual causal variants. The pipeline does not currently support the simulation of phenotypes from multiple causal variants.”

- It is not clear whether the results of phenotype-simulation based power estimation are based on quantitative or binary phenotype simulation. Should be added to methods and results section.

The results of phenotype-simulation power calculations are based on simulating binary phenotypes. We have clarified this point in the Methods and Results sections, and also in the footnotes of Figures and Tables.

Minor points:

- Line 33: Meaning unclear! Does it mean modifying the known relationship to fit different scenarios?

This is indeed what it means. We have extended this statement to specify that this is achieved “by modifying phenotype labels” (line 34).

- Line 108: powerBacGWAS does not predict phenotype labels so should it mean “recover the simulated causal markers”

We have edited this statement to indicate that: “We then perform GWAS to identify the sample size needed to recover the simulated genotype-phenotype relationship.” (line 111)

- Lines 229-230: meaning unclear! Should be rephrased.

**We have rephrased this statement to specify what we mean by “these variables”:
“Although it was expected that higher values of these variables (i.e. allele frequency, effect sizes, heritability and degree of homoplasy) would lead to an increase in power,” (line 269-271)**

- Lines 257-258: meaning unclear!

We wanted to stress that bacterial populations and the genotypes of individual strains were not in any case modified or simulated, but only phenotypes. We have rephrased this statement to make it clear: “In our approaches, only phenotypes were changed or simulated – bacterial genotypes and populations were not in any case simulated or modified.” (lines 310-131)

- Line 333: “Roary’s pan-genome analysis was not performed for the *M. tuberculosis* collection”. Explanation needed why this analysis and consequently pan-genome GWAS was not performed for *M. Tuberculosis*.

As indicated in the response to a previous comment, we have now applied the same genome analysis pipelines to all collections (lines 361-363), including the pan-genome analysis for the *M. tuberculosis* populations. Please note that we now employ Panaroo (<https://genomebiology.biomedcentral.com/articles/10.1186/s13059-020-02090-4>) for the calculation of pan-genomes, as it has been showed to outperform Roary.

- Figure 1: figure legend not readable!

We have now improved the quality of Figure 1.

- Figures 2 and 3: These figures will be more informative if the variations across 10 replications of data points are presented by error bars.

We have now added the option `--add_error_bars` to scripts *plot_gwas_runs.R* and *plot_gwas_runs_subsampling.R* to plot error bars. Please note that Figure 2 and Figure 3 now report power not average p-values, as requested by another reviewer.

Technical points:

powerBacGWAS tool is the main finding of the paper and can significantly improve the quality of future bacterial GWAS. The codes of powerBacGWAS are well-written and well-annotated, however, their implementation as a tool need two main improvements:

1) Workflow management: powerbacgwas produces lots of intermediate files which makes it difficult to use just by running commands one after another manually. Workflow management tools are developed to solve this problem and make the tool much more user-friendly by eliminating the need to manually handle the intermediates.

2) Parallelization: Although, powerBacGWAS parallelizes ancestral state construction, it does not parallelize the time-consuming process of all the GWAS runs which would make it difficult to use for power estimation in scenarios with lots of markers and large sample size.

I would recommend implementing powerBacGWAS as a workflow using management tools such as snakemake which can solve both handling of intermediates and parallelization of all the steps. I believe that developing a user-friendly version of the tool would significantly improve the impact of this paper as well. Nextflow, Airflow and other workflow management tools may also be good candidates.

We'd like to thank this reviewer for their implementation advise. Given the multiple software, library and package dependencies of the pipeline, we have now built a Docker image of the pipeline (<https://hub.docker.com/r/francesccoll/powerbacgwas>) to facilitate usage. We have additionally implemented a Nextflow pipeline (which uses this Docker image), to automate the multiple computational steps/scripts and parallelize the GWAS runs. Using the Nexflow implementation, the pipeline is reduced to only three steps: (1) preparation of input files, (2) ancestral state reconstruction and (3) GWAS runs. In the latter, the user can choose the type of variants, phenotype, GWAS method and approach to power calculations to be used. The GitHub page (<https://github.com/francesccoll/powerbacgwas>) now includes instructions on how to install PowerBacGWAS via Docker/Nextflow, or locally. The Usage wikpage (<https://github.com/francesccoll/powerbacgwas/wiki#usage>) now includes sections on how to run the pipeline via Docker/Nextflow ('Nextflow commands').

Minor points:

1. Following two dependencies are used by the tool but not mentioned in GitHub page. Need to be added to dependencies.

a. Perl

b. Bioperl (use for internal node annotation)

We have replaced the single Perl script (*annotate_nodes_newick.pl*) with a Python script (*annotate_nodes_newick.py*) performing the same task, to remove Perl dependencies.

2. "Three input files are needed to run the power calculations pipeline": Pan-genome GWAS and SNP-based GWAS are independent of each other (as shown in supplementary fig. 1) and users may need to just perform one. Explanation needed why both multi-sample VCF file and Roary-formatted pan-genome CSV file are needed to run powerBacGWAS.

The pan-genome GWAS and SNP-based GWAS are indeed independent of each other. We have edited this sections to make this point clear:

"The following input files are needed to run the PowerBacGWAS pipeline:

- **a phylogenetic tree in Newick format**
- **a multi-sample VCF file or a Roary-formatted pan-genome CSV file"**

3. In tutorial (<https://github.com/francesccoll/powerbacgwas/wiki#usage>), the subsampling approach, the command for running SNP-based approach is missing!

We added a new sub-section in the tutorial 'Applied to the detection of an isoniazid resistance mutation in *Mycobacterium tuberculosis* (VCF GWAS)' to include the sub-sampling SNP-based approach too.

4. In parameter file:

a. Mentioned: “NOTE: aphA gene observed causal variant odds ratio 89.10721003134796, thus 89 chosen as maximum” but maximum effect size is set to 100! Clarification needed!

This is an error message indicating that the maximum effect size set in the parameters file (100) is greater than the observed one (89.10). This is explained in the tutorial: “The script `_prepare_gwas_runs_subsampling_roary.py` will calculate the observed allele frequency and odds ratio of causal genes provided (`--causal-loci`), and will exit if the maximum allele frequency and effect sizes specified in the parameters file (`--parameters_file`) are greater than the observed ones.”

5. In parameter file used by ‘`prepare_gwas_runs.py`’ which includes phenotype-simulation based power estimation using GCTA, it is not clear how to differentiate between binary and quantitative phenotype simulation. In other words, how powerBacGWASim is set to calculate binary or quantitative power estimates?

We have now made `prepare_gwas_runs.py` and `prepare_gwas_runs_roary.py` scripts compatible with simulating quantitative phenotypes, and added option (`--phenotype_type`) to allow users to choose the type of phenotype to be simulated (“binary” or “quantitative” [Default: binary]).

6. In `simulate_binary_phenotype_vcf.py` line 377:

a. ‘`roary_samples_mut`’ and ‘`roary_samples_wt`’ are not defined before using! (probably a bug due to copying script lines from `simulate_binary_phenotype_roary.py` which leads the code to fail)

This bug has now been fixed.

Reviewer #2 (Remarks to the Author):

In their manuscript entitled “Alternative approaches to conduct power calculations for bacterial genome-wide association studies” Coll et al. present a new tool, PowerBacGWAS, which allows users to perform simulations using existing genomes to determine the effects of a range of input parameters on the ability to detect causal variants, and the effect sizes detectable with given sample sizes. The authors have implemented two simulation approaches, one which subsamples a dataset with a known phenotype, and the other simulates the phenotypes for a set of genomes. The utility of the tool is demonstrated by performing a large number of simulations testing the effect of a range of parameters including sample size, allele frequency, effect size, heritability and homoplasy in collections of three bacterial species. Using collections of existing genomes means that simulations are performed using genomes with real patterns of important parameters such as population structure and linkage disequilibrium. The pipeline is available on GitHub with a detailed tutorial.

PowerBacGWAS has the potential to be a useful tool to the bacterial GWAS community. However to improve its interpretability I think the tool should either add to or change the value reported from the simulations. The manuscript describes the tool as performing power calculations. Currently the mean p value per set of simulations is reported and presented in the manuscript, however this is not power, which is the ability to correctly reject the null

hypothesis, i.e. the proportion of the simulations in which the true/simulated causal hit is significant.

As recommended, PowerBacGWAS now calculates and reports power as the proportion of simulations in which the true/simulated causal hit is significant for a specific combination of parameters, in addition to the average p-value across replicate. See further details below.

More detail around which parameters changed and remained constant across the simulations would also improve understanding and comparison between the results.

We have now specified what parameters are kept constant and at what values in the footnotes of all figures and tables where power calculations results are presented. See further details below.

Detailed comments

1. The value reported is the average (I assume mean) p value for a set of simulations, but this doesn't tell you the power, the probability that the null hypothesis will be correctly rejected for a set of parameters. A more informative way to report the results would be to report the % of simulations where the causal variant is significant per set of parameters. Or for the tables, the number of genomes required for a set of parameters to give a specified level of power.

As recommended, PowerBacGWAS now calculates and reports power as the proportion of simulations in which the p-value of causal variants is significant per set of parameters. All figures in the manuscript now report power in the y-axis as a function of sample sizes (x-axis) and other parameters. Tables in the manuscript now report the sample sizes required for a set of parameters to achieve 80% power. Figures have been updated on the tutorial (<https://github.com/francescoll/powerbacgwas/wiki#usage>).

a. As the results are currently reported, some sets of parameters are reported as 'non-detectable using GWAS', however the mean p value of the causal variant being below the significance threshold does not necessarily mean that power was zero.

PowerBacGWAS final plots now report the power (%) in the y-axis as a function of sample size, allele frequency and effect size or heritability.

b. Likewise where it is reported that a given sample size is enough to detect a variant at a particular effect size, this is true in that the results mean that power is non-zero, but it does not tell you power itself. Users may want to know the increase in power for an increase in sample size above the level at which an effect is detectable. Power is more informative on study design than the mean p value attained by simulations.

See responses above.

2. It would also be very useful if the false positive rate could also be reported for the simulations alongside power. Or a statement made on whether a low FPR is maintained as power increases with a given parameter.

We agree that reporting false positive rates would be very informative, however, it was out of the scope of this work to investigate false positives. As pointed by other authors,

false positives in bacterial GWAS may arise from a variety of sources, including the degree of missingness of variants, sequencing batch effects, regions in the genome that are hard to genotype, and the degree of stratification and linkage disequilibrium of causal variants in the population. We believe this question warrants further investigation and should, on its own, be the focus of a future study that describes the factors giving rise to false positives, and how to best identify and minimize them. We have now added this point to the Discussion (lines 302-307).

3. The authors describe the results of a range of different parameters that impact the power to detect causal variants. However when a parameter is changed, it needs to be clearer what the values are for the other parameters, and if they are kept constant while the parameter of interest is being changed. In particular, when the effect size is being assessed, is heritability kept constant and at what value? When heritability is being assessed, is the causal variant effect size kept constant and at what value?

PowerBacGWAS pipeline is designed to vary every single parameter while keeping the rest constant. The range of values each parameter is allowed to vary along is specified in the parameters file (see <https://github.com/francescoll/powerbacgwas/wiki#usage> for examples). We have now been more explicit about what parameters are kept constant and at what values in the footnotes of all figures and tables where power calculations results are presented.

When using the our custom approach to simulate binary phenotypes, full heritability is assumed. As requested by another reviewer, we now describe the full set of equations used to simulate binary phenotypes for a given odds ratio. When using the phenotype-simulation approach, the pipeline allows to vary the effect sizes of causal variants or their heritability, but not both at the same time. If multiple values of effect sizes and heritability are specified in the parameters file, the script `prepare_gwas_runs.py` will exit with an error.

4. I think the language around the impact of variant frequency needs to be clearer, it is not the frequency of a particular variant but the minor allele frequency of either presence/absence of the variant that determines power. The language in the manuscript seems to suggest that the higher the frequency the higher the power E.g. line 144.

We have now used the term MAF (minor allele frequency) instead of the previously used terms “genotype frequency” or “allele frequency” for clarity.

a. On this note, Supplementary Table 1 legend states that the complete *E. faecium* single-clade collection was not used because the resistance conferring gene was 79% frequent, too high to perform power calculations. It is not clear why this is too high.

We found that if the MAF is too high (i.e. well above 50%) it may not be possible to sub-sample enough wild-type samples (i.e. isolates without the causal allele) to simulate binary phenotypes for a given odds ratio, that is, to resolved the full set of equations implemented in our method.

5. Burden testing

a. How were the burden tests performed?

We've added a NOTE in the wikipage (<https://github.com/francescoll/powerbacgwas/wiki>) clarifying this:

NOTE: Burden tests are performed as implemented by PySeer, regardless of the annotation of VCF alleles (that is, their predicted functional effect). It is often recommended to group variants with the same predicted functional effect (e.g. loss of function mutations within a gene). In this case, you will need to filter the input VCF file as explained in [PySeer's manual] (<https://pyseer.readthedocs.io/en/master/usage.html#rare-variants>).

b. Line 179: 'could detect variants of down to 2.5% frequency' what does variant mean in this case? Is this discussing a joint burden of mutations that together add up to 2.5% of genomes affected, or does this mean it can detect a gene containing multiple variants, the lowest of which is individually found at 2.5% frequency?

We have rephrased this statement to make this point clear: "and could detect mutated genes down to 2.5% MAF, not detectable by a SNP GWAS (Table 3). Here, the MAF of genes in a burden test refers to the percentage of samples carrying one or multiple mutations in the same gene."

c. Line 177-178: It is not the case for all parameter combinations in Table 3 (where effect size is adjusted) that the number of genomes required is smaller for the burden GWAS.

This is indeed the case. We've deleted the statements below from this paragraph for simplicity as, generally speaking, the number of genomes required is either the same or more often smaller for the burden GWAS.

"The sample sizes required to detect SNPs of the same effect size and MAF with 80% power varied by population, although they were lower when using burden testing."
(lines 198 - 201)

d. Supplementary Table 3 (where heritability is adjusted) the burden GWAS is sometimes able to detect effects at lower frequencies than the variant GWAS is able to, but where both are able to detect an effect at a particular frequency, the burden GWAS often requires a much larger number of genomes which feels contradictory, could this be explained? Is the simulated effect size being kept constant for these simulations?

The effect size is kept constant for these simulations. We have now updated Supplementary Table 3 with the sample sizes required to achieve 80% power.

6. Line 180-181: A larger number of genomes are required to detect the same odds ratio for SNPs versus genes, is this due to the higher multiple testing correction threshold or do the authors think there is another reason for this?

The higher multiple testing correction for SNPs compared to genes should be a factor. Also, the fact that genes may contain multiple SNPs, and these can originate in different parts of the tree, makes causal genes more homoplastic, and thus easier to identify using GWAS, than individual SNPs.

7. What is the kinship matrix for the LMM calculated from? Is the kinship matrix kept the same for the SNP versus gene analysis per species?

The kinship matrix for LMM is calculated from the phylogenetic tree using PySeer's script *phylogeny_distance.py* and thus is kept the same for all GWAS analyses (SNP, burden testing and pan-genome GWAS) applied to the same population. We've added this point to the Methods section.

8. Line 442-444: More detail is required on the method to simulate binary phenotypes and the differences to the GCTA phenotype simulations explained. Why does heritability not have such a strong effect when using this method? Is heritability kept constant through these simulations, if so what value is it set to?

We now describe the full set of equations our custom method uses to simulate binary phenotypes for a given odds ratio, implemented in the scripts *simulate_binary_phenotype_vcf.py* and *simulate_binary_phenotype_roary.py*. See lines 558-564. When using the our custom approach to simulate binary phenotypes, full heritability is assumed.

9. Figure 3: Could the authors explain why the mean p values for TB at 50/100 homoplasy steps oscillate so much in significance with changing sample size compared to the other simulations?

Mean p-values (now power) in Figure 3E for SNPs with 50-100 homoplasy steps is particularly noisy due to the low number of SNPs in this population (*M. tuberculosis* species-wide population) arising 50 to 100 times in the phylogeny (only 9 variants), which makes average p-values (now power) of such a small sample to fluctuate a lot. Many more variants could be randomly sampled with 10-50 homoplasy steps (n=70), 5-10 (n=381) and 1-5 steps (n=468), making the p-value averages of such larger samples more stable. We added this explanation to the footnote of Figure 3.

10. It appears that multiple causal variants can be simulated. In the case where more than one causal variant is simulated, are these modelled additively? When the variants are tested individually (not by a burden test) how does the pipeline report the result of the simulation?

Although the parameters file includes a variable named *number_causal_variants* (set to 1 in all simulations), the pipeline does not support multiple causal variants (i.e. multiple acquired genes or multiple mutated loci) to be used for simulating phenotypes, as this would require a genetic model to be chosen too. We had made this point in the Discussion ("Further work is needed to perform power calculations for complex bacterial phenotypes involving multiple loci (e.g. epistatic effects)."), and now added a NOTE in the manual (GitHub wikipage) of the project too, to make this point clear.

11. Samples and SNP sites with an excess of 10% missing calls are removed, but how are the remaining missing calls handled in the GWAS? If missing calls are ignored when a site is tested then the simulations will be comparing different numbers of genomes depending on the call rate of the causal variants selected.

Please note that now, for consistency, we have applied the same genome analysis and phylogenetic tree reconstruction to all collections, to avoid the variation in power estimates that may arise from employing different genome analysis pipelines. The

Methods section ‘Genome analysis pipelines’ has been rewritten to reflect these changes.

The rate of missing calls is indeed ignored when randomly choosing SNPs to simulate phenotypes. However, this step is repeated multiple times (specified by the variable *sampling_repetitions* in the parameters file) which means that the power or average p-values calculated across simulations will account for the multiple and most common missing call rates present in the population.

Minor comments

12. Line 108: Should this say ‘identify the sample size needed to recover the simulated causal variant’ instead of ‘identify the sample size needed to recover the simulated phenotype’

We have edited this statement to: “We then perform GWAS to identify the sample size needed to recover the simulated genotype-phenotype relationship.”

13. Line 130, 131 and 133: Are these the allele frequencies in the whole population or in just those phenotyped for the antibiotic in question, i.e. is this the maximum frequency that could be tested in the phenotyped dataset or was that lower?

The allele frequencies are those calculated in the whole population. We have added a new footnote in Table 1 indicating this: “⁵The MAF was calculated in the whole population not in just the samples phenotyped for the antibiotic in question.”

14. Line 485-486: How were the number of independent tests defined for the Bonferroni correction?

The number of independent tests is calculated as the number of unique SNP allele patterns (for variant GWAS) and number of unique gene presence/absence patterns (for pan-genome GWAS) across all samples. We explain how this can be calculated from VCF and pan-genome files in the manual.

15. Line 176-177: There is a supplementary figure for the burden test results but not for the SNP GWAS results, could a figure be added for the SNP results?

We have now added a new figure (Supplementary Figure 4) with the SNP results.

16. Line 213: It will also require knowing or making an assumption on the heritability of the phenotype

We’ve changed this statement to: “Either way, conducting power calculations will require making a set of assumptions as to the type of causal genotypes (i.e. caused by the acquisition of genes or mutations), their MAF, effect sizes and heritability.”

17. The methods need to state number of repeats per set of simulation parameters

We have now added this in the Methods:

“The number of times this variant sampling step is repeated can be chosen in the parameters file. This was set to 10 per set of simulation parameters in the analyses

presented in this manuscript.” And “The number of times this phenotype simulation step is repeated can be chosen in the parameters file. This was set to 10 per set of simulation parameters in the analyses presented in this manuscript.”

Reviewer #3 (Remarks to the Author):

In this paper, Coll and colleagues introduces PowerBacGWAS, a bioinformatic approach to estimate statistical power in bacterial GWAS. This is an interesting topic and it is easy to see the potential need for such a tool in the field. However, I have some concerns about the statistical rigor of this study which I will illustrate below.

Major comments:

1. The authors seem to have misunderstood the concept of statistical power, which is unfortunate for a paper about power calculation. Given a type-I error rate (e.g., 0.05, or a Bonferroni-corrected threshold), statistical power is the probability for a test to give a p-value below this cutoff under the alternative hypothesis. Therefore, if we fix the level of statistical power we want (e.g., 80%), needed sample size can be calculated. Similarly, if we fix the sample size (along with effect size and other factors), power can be estimated as a percentage between 0 and 1. This is not how this study approached this problem. Using the proposed "subsampling approach" as an example, the authors basically started from a known genotype-phenotype association with large N, then reduced N (let's assume the effect size to be fixed for simplicity), and calculated one p-value for each corresponding N. If the p-value reaches the significance threshold, the authors claim to have had enough power.

This approach completely ignored the fact that p-value itself is a random variable and if the study is repeated in an independent dataset with all the identical parameters, p will be different and may or may not be significant. Although in some simple scenarios, this issue may be resolved by simply repeating the subsampling procedure many times to get an empirical distribution of p-values, it gets trickier if the subsampled N is relatively large since different repeats will have substantial sample overlap and therefore the p-values obtained from these repeats won't be independent.

We thank this reviewer for raising this important concern, which was also raised by Reviewer 2. To address this recommendation, PowerBacGWAS now calculates and reports power as the proportion of simulations in which the true/simulated causal hit is significant for a specific combination of parameters, in addition to the average p-value we had previously reported. Please note that the power calculation plots produced by the pipeline (also the ones included in the Figures of this manuscript) now report power in the y-axis as a function of sample size (in the x-axis), minor allele frequency, and effect size or heritability. Tables in the manuscript now report sample sizes required to achieve an 80% power, for any set of parameters indicated.

2. Throughout the Methods section, the authors focused on describing the bioinformatic pipeline, mentioning numerous pieces of scripts. However, no statistical models were presented. What is the linear mixed model used to conduct GWAS in this study? What are the fixed/random effects in the LMM? If the GWAS approach is just a LMM, why can't general power calculation tools designed for LMM be used to estimate power for bacterial GWAS? How is bacterial GWAS special compared to any general LMM application? The authors claimed that power calculators for human GWAS cannot be applied due to unique characteristics of bacterial populations (clonal reproduction, strong population structure, and

uneven and varying degrees of recombination), but it is unclear why this is the case without showing the model.

We omitted describing the statistical method (i.e. LMM) implemented by the GWAS software tool used here (i.e. PySeer) as this is explained in the original publication (<https://academic.oup.com/bioinformatics/article/34/24/4310/5047751>). The LMM method implemented in PySeer uses a kinship matrix to estimate random effects which is used to control for population structure. The kinship matrix of relations between all pairs of samples is derived from a phylogenetic tree. The consequence of this is that the phylogenetic relationships of bacterial strains in the population of study will be a factor influencing the power to detect causal variants. This is why we did not use general power calculation tools designed for LMM to estimate power for bacterial GWAS, and instead undertook a more empiric approach by using real bacterial populations and genomes, as explained in the last paragraph of the Discussion.

3. GCTA was used to simulate phenotypes in the proposed "phenotype simulation approach". GCTA was designed to model highly polygenic complex traits with random genetic effects equally attributed to all variants in the model. Is this compatible with the genetic architecture of bacteria phenotypes? Although technically it is possible to specify the causal SNP effect sizes and simply use GCTA to obtain the error term, some details are still lacking such as how many causal SNPs are there and if they all have the same pre-specified effect sizes?

We used a single causal variant to simulate phenotypes, for simplicity and to facilitate interpretation. This is not to say that bacterial phenotypes are only caused by single variants (although this is often the case for antibiotic resistance phenotypes) but to assess the limit of detection of individual causal variants, which is often the unit of interpretation. The pipeline does not currently support multiple causal variants to be used for simulating phenotypes, as this would require a genetic model to be chosen too. We had made this point in the Discussion ("Further work is needed to perform power calculations for complex bacterial phenotypes involving multiple loci (e.g. epistatic effects)."), and have now been more explicit in the Methods section: "We used a single causal variant (i.e. single acquired gene in the pan-genome GWAS; single SNP in the SNP GWAS and single mutated gene in the burden GWAS) to simulate phenotypes, to assess the limit of detection of individual causal variants. The pipeline does not currently support the simulation of phenotypes from multiple causal variants."

4. The section about simulating binary phenotypes is a bit odd. Prevalence of binary traits was not mentioned in the paper. Related to this, when heritability was specified in the analysis, is it on the liability scale or the observed scale? It also isn't clear why variants' effect size (odds ratio) didn't matter any more once the heritability was specified. The way it was described in the Methods section was more like an observation than an explanation. In the same paragraph on page 18, it was stated that "this method solves a set of equations to find the number of cases and controls..." and the equations were not shown/explained. Overall, the Methods section (which is what I focused on the most since this is a methodology paper) reads more like a user manual that explains which scripts the users should apply instead of a manuscript's Methods section that explains the statistical and computational details. This needs to be improved.

We have added more text in the Methods section explaining our custom approach to simulate binary phenotypes, and the set of mathematical equations that are solved in this script. See lines 554-576.

5. It is unclear why the allele frequencies were specified differently between panels A and B of Figure 2. Also, is it reasonable to specify similar parameter values (for both allele frequency and effect size) in panel C which is about burden tests? By the way, panel C is mislabeled as "A" in the figure.

The maximum allele frequency that could be tested with the sub-sampling approach (the approach using known AMR genotype-phenotype relationships) is determined by the maximum allele frequency observed of known causal AMR variants (shown in the last column of Table 1). For simplicity, we have now kept only 1%, 2.5%, 5% and 10% MAF in all three plots. The effect sizes tested are the same across all three plots: 1.5, 5, 10 and 100. We've corrected the label of Figure 2C.

6. Acronym "AMR" isn't defined in the paper.

This has now been fixed: "Next, we searched for a known antimicrobial resistance (AMR) phenotype-genotype relationship in each population..."

7. Causal language was used when explaining GWAS effect sizes (e.g., page 4, line 85). Is bacterial GWAS not aiming to identify indirectly associated markers in LD with the (potentially unobserved) causal variants?

In the context of our approach to power calculations, in which known AMR causal variants are used (sub-sampling approach) or specific genetic variants are randomly picked as "causal variants" to simulate phenotypes (phenotype simulation approach), we think we can use the term causal variants. In a more general context, causal variants may not be the right term to use, as GWAS will detect variants in LD with other markers associated with a phenotype, not necessarily only the causal variants.

Reviewers' comments:

Reviewer #2 (Remarks to the Author):

The manuscript has been revised to now report power for the simulation results instead of mean p-value, which greatly improves the manuscript and the tool PowerBacGWAS. Most of the reviewers comments have been addressed, however I believe there are some remaining points to be clarified.

1. The authors have now added that 10 simulations were performed per set of parameters, so it is unclear why the power estimates in the figures (% of the 10 simulations where the causal variant was significant) are not all a multiple of 10%. Should this be 100 simulations?
2. The term "AF" (allele frequency) has been replaced with "MAF" (minor allele frequency) throughout, but this has caused a misunderstanding of the definition of MAF. MAF is the frequency of the minor allele, which may be presence or may be absence of a particular variant. E.g. in line 134, the MAF cannot be 56.3%, that is the frequency of the allele of interest, but the frequency of the minor allele would be 43.7%. This also applies to two of the MAFs in Table 1. Likewise, line 496 "MAF (the proportion of samples carrying causal minor allele)" the causal allele may not be the minor allele.
3. Can it be clarified whether the MAF of the causal variant in the simulations is the MAF in the subset of phenotyped and called isolates (not the full dataset and not including isolates with missing calls at the site). In the authors response they state that missing calls are not dealt with in the simulations and up to 10% missingness is allowed, does this change the actual sample size used in the simulations by up to 10% compared to the sample size reported? The parameter important for power in the GWAS will be the MAF in the phenotyped and called isolates at the site, so it needs to be clearer how this is defined and the actual sample sizes tested.
4. Supplementary Table 3 is not referenced in the text. As in my original review, it is still unclear why the burden GWAS sometimes reaches 80% power at lower MAFs than the variant GWAS, but where both reach 80% power at a particular MAF, the burden GWAS sometimes requires a much larger number of genomes. This is contradictory with the text (lines 195-197) stating that the burden testing had more power than the SNP GWAS. Could these lines be amended to discuss the results in this table.
5. The authors response states that they have specified which parameters are kept constant and at what values in the footnotes of all figures and tables, but this is still missing for most figures and tables, can this please be added. One example is supp table 2, what was the simulated effect size? It isn't in the text or table legend.
6. Table 1 – is it correct that the median R2 for *E. faecium* is lower for the single clade than species-wide? Typo for *K. pneumoniae* species-wide IQR.
7. The points on the new figure 2 are difficult to distinguish. For example, in panel A it's not possible to tell which of the red (allele freq 0.01) increases to 100% power at 1000 samples. Could these be made clearer, perhaps with different line types.
8. Supplementary figure 4 not referenced in text

Reviewer #4 (Remarks to the Author):

The authors have done a great deal of work in responding to each comment in a rigorous manner. This has resulted in the manuscript being significantly improved, the software being more usable and I anticipate the package being widely adopted.

Referee expertise:

Referee #2: population genetics, microbial genomics

Referee #4: Bacterial GWAS

Reviewers' comments:

Reviewer #2 (Remarks to the Author):

The manuscript has been revised to now report power for the simulation results instead of mean p-value, which greatly improves the manuscript and the tool PowerBacGWAS. Most of the reviewers comments have been addressed, however I believe there are some remaining points to be clarified.

1. The authors have now added that 10 simulations were performed per set of parameters, so it is unclear why the power estimates in the figures (% of the 10 simulations where the causal variant was significant) are not all a multiple of 10%. Should this be 100 simulations?

The total number of simulations depends on two values that can be chosen by the user: the number of times the “variant sampling step” is repeated, which is set to 10 (lines 421-423), and the number of times the “phenotype simulation step” is repeated (lines 488-490), which is set to 10. Thus, for each set of parameters, a total of 100 simulations is performed, from which power is estimated.

We have added the following statement in lines 502-506: “For each set of parameters, the total number of GWAS replicates is determined by the number of times the “variant sampling step” is repeated multiplied by the number of times the “phenotype simulation step” is repeated.”

2. The term “AF” (allele frequency) has been replaced with “MAF” (minor allele frequency) throughout, but this has caused a misunderstanding of the definition of MAF. MAF is the frequency of the minor allele, which may be presence or may be absence of a particular variant. E.g. in line 134, the MAF cannot be 56.3%, that is the frequency of the allele of interest, but the frequency of the minor allele would be 43.7%. This also applies to two of the MAFs in Table 1. Likewise, line 496 “MAF (the proportion of samples carrying causal minor allele)” the causal allele may not be the minor allele.

We’ve brought back the term “allele frequency (AF)” in the context of the sub-sampling approach, that is when we are referring to the frequency of a specific allele of interest (i.e. the causal allele), which may or may not be the minor allele. In the context of a pan-genome GWAS, we now refer to gene frequencies (proportion of the population carrying a particular gene). In the context of variant and burden GWAS, when variants are randomly chosen based on their minor allele frequency, we’ve kept the term MAF.

3. Can it be clarified whether the MAF of the causal variant in the simulations is the MAF in the subset of phenotyped and called isolates (not the full dataset and not including isolates with missing calls at the site). In the authors response they state that missing calls are not dealt with in the simulations and up to 10% missingness is allowed, does this change the

actual sample size used in the simulations by up to 10% compared to the sample size reported? The parameter important for power in the GWAS will be the MAF in the phenotyped and called isolates at the site, so it needs to be clearer how this is defined and the actual sample sizes tested.

MAF is calculated in script *sample_causal_variants_from_vcf.py* by dividing the number of isolates with the minor allele by the total number of isolates in the dataset, not only the ones with called alleles at the site. We agree we shouldn't have counted isolates with missing calls at the site when calculating MAF, and we will implement the proposed MAF calculation in the next version of the pipeline. In the context of the results presented in this manuscript, given that multiple variants are randomly chosen per set of parameters (specifically 10 as indicated in comment 1), we believe this had little effect on power estimates.

4. Supplementary Table 3 is not referenced in the text.

Supplementary Table 3 is now referenced in lines 199-201: "As for the detection of causal acquired genes in a pan-genome GWAS, higher heritability resulted in more power to detect common variants (Supplementary Table 3) but had little effect on the detection of rarer variants."

As in my original review, it is still unclear why the burden GWAS sometimes reaches 80% power at lower MAFs than the variant GWAS, but where both reach 80% power at a particular MAF, the burden GWAS sometimes requires a much larger number of genomes. This is contradictory with the text (lines 195-197) stating that the burden testing had more power than the SNP GWAS. Could these lines be amended to discuss the results in this table.

We have investigated this point further. First, we have re-run the power calculations for the burden GWAS with varying effect sizes (results updated in Table 3). These results show that the burden GWAS reaches 80% power at lower MAFs than the variant GWAS, and when both reach 80% power at a particular MAF, the burden GWAS requires a similar, or more often a lower, number of genomes. This is the case for all populations studied (Table 3) except for the *M. tuberculosis* single-clade population.

However, the power calculations results for the variant and burden GWAS with varying heritability values (Supplementary Table 3) show indeed that the burden GWAS sometimes requires a much larger number of genomes at a particular MAF, which contradicts the statement in lines 195-197 (now lines 197-199).

This difference is most likely due to how GCTA simulates phenotypes. Here GCTA is used to simulate phenotypes from heritability values (Supplementary Table 2 and 3) but not from effect sizes. GCTA models phenotypes with random genetic effects equally attributed to all variants. This means that when considering multiple SNPs in the same gene, the overall heritability is distributed among these multiple SNPs. This would explain why the burden GWAS sometimes requires a much larger number of genomes than the variant GWAS, where the whole heritability is attributable to a single SNP. On the contrary, when using our custom method (Table 3), we treat all mutated versions of the same gene as having the same effect size when simulating binary phenotypes for burden GWAS.

As indicated in the Methods section: “We used a single causal variant (i.e. single acquired gene in the pan-genome GWAS; single SNP in the SNP GWAS and single mutated gene in the burden GWAS) to simulate phenotypes, to assess the limit of detection of individual causal variants. The pipeline does not currently support the simulation of phenotypes from multiple causal variants.”

We have thus decided not to present the results of using GCTA to simulate phenotypes from multiple variants (we only present GCTA phenotype simulations from single variants, in Supplementary Table 2 and 3), as we haven’t investigated to what extent the multi-variant genetic models used by GCTA are compatible with the genetic architecture of bacteria phenotypes.

10. It appears that multiple causal variants can be simulated. In the case where more than one causal variant is simulated, are these modelled additively? When the variants are tested individually (not by a burden test) how does the pipeline report the result of the simulation?

Although the parameters file includes a variable named *number_causal_variants* (set to 1 in all simulations), the pipeline does not currently support multiple causal variants (i.e. multiple acquired genes or multiple mutated loci) to be used for simulating phenotypes, as this would require a genetic model to be chosen too. We had made this point in the Discussion (“Further work is needed to perform power calculations for complex bacterial phenotypes involving multiple loci (e.g. epistatic effects).”), and now added a NOTE in the manual (GitHub wikpage) of the project too, to make this point clear.

We have now been more explicit in the Methods section: “We used a single causal variant (i.e. single acquired gene in the pan-genome GWAS; single SNP in the SNP GWAS and single mutated gene in the burden GWAS) to simulate phenotypes, to assess the limit of detection of individual causal variants. The pipeline does not currently support the simulation of phenotypes from multiple causal variants.”

5. The authors response states that they have specified which parameters are kept constant and at what values in the footnotes of all figures and tables, but this is still missing for most figures and tables, can this please be added. One example is supp table 2, what was the simulated effect size? It isn’t in the text or table legend.

We have now indicated what parameters are kept constant in the footnotes of all figures and tables. Specifically, in table/figures showing varying effect sizes we’ve indicated that “full heritability is assumed”; and in table/figures showing varying heritability values that “effect size was kept constant at an odds ratio of 2”.

6. Table 1 – is it correct that the median R2 for *E. faecium* is lower for the single clade than species-wide? Typo for *K. pneumoniae* species-wide IQR.

It is correct that the median R2 for the *E. faecium* clade population is lower than that for the species-wide population. The *E. faecium* clade population is considerably more diverse (see average diversity, third column, calculated as the mean pairwise genetic distance between isolates and expressed as number of SNPs per kilobase) than the *M. tuberculosis* and *K. pneumoniae* clade populations. We have corrected the typo in the IQR for *K. pneumoniae* species-wide. The correct IQR is: 0.67 (0.37 - 1.00).

7. The points on the new figure 2 are difficult to distinguish. For example, in panel A it's not possible to tell which of the red (allele freq 0.01) increases to 100% power at 1000 samples. Could these be made clearer, perhaps with different line types.

We have improved Figure 2 to indicate different effect sizes with different line types, as recommended. Now it's possible to tell that the red line achieving 100% power at 1000 samples is that of effect size (odds ratio) of 100 (i.e. solid line).

8. Supplementary figure 4 not referenced in text

**Supplementary Figure 4 is referenced in the footnote of Table 3 (lines 610-611):
“Supplementary Figure 3 and 4 show the PowerBacGWAS plots from which the results in this table were extracted from.”**

Reviewer #4 (Remarks to the Author):

The authors have done a great deal of work in responding to each comment in a rigorous manner. This has resulted in the manuscript being significantly improved, the software being more usable and I anticipate the package being widely adopted.

We thank Reviewer #4 for assessing all comments in this rebuttal.

** See the Nature Portfolio author and referees' website at www.nature.com/authors for information about policies, services and author benefits

Communications Biology is committed to improving transparency in authorship. As part of our efforts in this direction, we are now requesting that all authors identified as 'corresponding author' create and link their Open Researcher and Contributor Identifier (ORCID) with their account on the Manuscript Tracking System prior to acceptance. ORCID helps the scientific community achieve unambiguous attribution of all scholarly contributions. You can create and link your ORCID from the home page of the Manuscript Tracking System by clicking on 'Modify my Springer Nature account' and following the instructions in the link below. Please also inform all co-authors that they can add their ORCIDs to their accounts and that they must do so prior to acceptance.

If you experience problems in linking your ORCID, please contact the Platform Support Helpdesk.

Our flexible approach during the COVID-19 pandemic

If you need more time at any stage of the peer-review process, please do let us know. While our systems will continue to remind you of the original timelines, we aim to be as flexible as possible during the current pandemic.

COMMSBIO - This email has been sent through the Springer Nature Tracking System NY-610A-NPG&MTS

Confidentiality Statement:

This e-mail is confidential and subject to copyright. Any unauthorised use or disclosure of its contents is prohibited. If you have received this email in error please notify our Manuscript Tracking System Helpdesk team at <http://platformsupport.nature.com> .

Details of the confidentiality and pre-publicity policy may be found here <http://www.nature.com/authors/policies/confidentiality.html>

Privacy Policy | Update Profile

REVIEWERS' COMMENTS:

Reviewer #5 (Remarks to the Author):

This is a valuable manuscript and method that is well-implemented using nextflow and the useful wiki page.

The points 1, 2, 4 from Reviewer #3 are adequately addressed.

The remaining comment 3 is not completely addressed because the authors propose to postpone implementation of the proposed MAF calculation and evaluation how the difference in MAF calculation has effect on power calculation. I agree with the authors that the effect would be slight, but currently they don't present any data to support it. I'm not sure if you and the reviewer are satisfied with the answer for this journal. Also, I cannot find the description regarding "up to 10% missingness is allowed" in the main text or in the wiki.

Please consider to incorporate following comments for further improvement.

First of all, this method is applicable to any bacterial population that has a clonal phylogeny enabling ancestral state reconstruction, which is not possible if recombination rate is very high for example in *Helicobacter pylori*. I recommend to modify the last sentence in Abstract and descriptions regarding limitation of this study in Discussion.

L107: minimum allele frequency -> minor allele frequency

L168-170: 500 to 600 genomes are not necessarily needed according to Table 2, for example 400 genomes is needed for the two datasets of *E. faecium* when MAF is 10% and odds ratio is 10.

L185-187: Supplementary Table 3 shows results of SNP-GWAS, rather than pan-genome GWAS, according to the title, which is not consistent with the description in L185-187.

L202-204: Could you add a discussion regarding why higher sample sizes were required in the single-clade population in the case of *K. pneumoniae*?

L448 "the effect size (odds ratio)": when a quantitative phenotype is specified in the phenotype simulation approach, the effect size is not odds ratio. Please clarify definition of the effect size when a quantitative phenotype is specified in the phenotype simulation approach.

Burden testing GWAS: I recommend to add "PowerBacGWAS can also be used to conduct power calculations for burden test GWAS, where VCF variants are aggregated within chromosomal regions (e.g. genes), and regions tested for association." (currently only written in the wiki) to somewhere in the manuscript.

powerbacgwas program: according to the wiki, I executed the following command in a computing cluster "nextflow run nextflow/main.nf --outprefix kpn_clade.var --tree ./data/kpn_clade.tree.annotated.nwk --vcf ./data/kpn_clade.vcf.gz --parameters ./data/kpn_clade.parameters.binary.efs.txt --run_phensim_vcf --num_cores 8 --gwas_tmp_dir tmp/gwas_tmp_kpn_clade_var_efs/", which is taking more than a whole day and has not yet finished. I suggest to write usual computation time in the wiki or manuscript for users to understand it in advance.

REVIEWERS' COMMENTS:

Reviewer #5 (Remarks to the Author):

This is a valuable manuscript and method that is well-implemented using nextflow and the useful wiki page.

The points 1, 2, 4 from Reviewer #3 are adequately addressed.

The remaining comment 3 is not completely addressed because the authors propose to postpone implementation of the proposed MAF calculation and evaluation how the difference in MAF calculation has effect on power calculation. I agree with the authors that the effect would be slight, but currently they don't present any data to support it. I'm not sure if you and the reviewer are satisfied with the answer for this journal. Also, I cannot find the description regarding "up to 10% missingness is allowed" in the main text or in the wiki.

We have amended the script *sample_casual_variants_from_vcf.py* to calculate the allele frequency not including isolates with missing calls at the site, as recommended by Reviewer #2, and updated the Docker images accordingly.

The description of "up to 10% missingness is allowed" applies to a previous version of the manuscript wherein a different variant calling pipeline had been used. In this latest version, we run *Snippy* for all datasets as this tool is recommended as a general purpose bacterial SNP-calling pipeline (Bush JB *et al.* 2020. DOI: 10.1093/gigascience/giaa007) and has been shown to minimize false positives calls (Bush JB *et al.* 2021. DOI: 10.1099/mgen.0.000615). However, a limitation of *Snippy* is that missing SNP calls are not retained in the output files used (*.consensus.subs.fa*, see <https://github.com/tseemann/snippy>). We have mentioned this limitation under the Methods section 'Genome analysis pipelines' (lines 360-361). Given the lack of missing calls in the multi-sample VCF files used, we cannot evaluate the difference in MAF calculation has effect on power calculation.

Please consider to incorporate following comments for further improvement.

First of all, this method is applicable to any bacterial population that has a clonal phylogeny enabling ancestral state reconstruction, which is not possible if recombination rate is very high for example in *Helicobacter pylori*. I recommend to modify the last sentence in Abstract and descriptions regarding limitation of this study in Discussion.

We've edited the Abstract to indicate that the pipeline "and can be applied to other bacterial populations." instead of "to any bacterial population" and also added the following statement to the limitations paragraph in the Discussion: "A requirement to run the *phenotype-simulation* approach is that the ancestral state of genetic variants can be reconstructed so that these can be selected based on their degree of homoplasy. This prerequisite may not be possible for bacterial populations with high recombination rates." (lines 280-283)

L107: minimum allele frequency -> minor allele frequency

This has been corrected. All other instances of this term have the right spelling or are abbreviated as MAF.

L168-170: 500 to 600 genomes are not necessarily needed according to Table 2, for example 400 genomes is needed for the two datasets of *E. faecium* when MAF is 10% and odds ratio is 10.

This is correct. Here we wanted to indicate that a minimum of 500 to 600 genomes are needed to detect acquired genes of 10% frequency starting with moderate effect sizes (OR=5). Although, as pointed, required sample sizes become smaller with greater effect sizes (OR=10). We have edited this statement to indicate: “Specifically, a minimum of 500 to 600 genomes were needed...” (line 181)

L185-187: Supplementary Table 3 shows results of SNP-GWAS, rather than pan-genome GWAS, according to the title, which is not consistent with the description in L185-187.

Supplementary Table 3 is the right table here. The previous wording “As for the detection of causal acquired genes in a pan-genome GWAS,” was confusing, which should have been: “as it was the case for the detection of causal acquired genes...”. We have changed this statement to make this clear (lines 200-203).

L202-204: Could you add a discussion regarding why higher sample sizes were required in the single-clade population in the case of *K. pneumoniae*?

We have added the following statement to the Discussion (lines 259-262):

“Here we show that MAF, homoplasy and effect size of causal SNPs all have a measurable effect in the sample sizes required to detect them. The fact that the magnitude of such effect varies by population points to population-specific factors having an influence too.” This statements are followed by “our recommendation to conduct power calculations using real genomes for the specific bacterial population of study”.

L448 “the effect size (odds ratio)”: when a quantitative phenotype is specified in the phenotype simulation approach, the effect size is not odds ratio. Please clarify definition of the effect size when a quantitative phenotype is specified in the phenotype simulation approach.

We have clarified this point in the GitHub Wiki page: “The units of effect_size must be specified as odds ratio (where 0 to 1 indicate negative association, 1 no association, and > 1 positive association) when simulating a binary phenotype, or in beta units when simulating a quantitative phenotype.” and in the Methods section lines 469-470.

Burden testing GWAS: I recommend to add “PowerBacGWAS can also be used to conduct power calculations for burden test GWAS, where VCF variants are aggregated within chromosomal regions (e.g. genes), and regions tested for association.” (currently only written in the wiki) to somewhere in the manuscript.

As recommended we made this point more explicit in the manuscript by adding statements in the Abstract in line 37 (“that supports power calculations for burden testing, pan-genome and variant GWAS”) and Methods in lines 422-429 (“It supports power calculations for acquired genes (pan-genome GWAS), VCF variants (variant GWAS) and burden testing, where VCF variants are aggregated within chromosomal regions (e.g. genes), and regions tested for association.”) and in lines 458-459 (“this approach consists in sampling existing genes from the pan-genome, variants from a VCF file or mutated regions (i.e. for burden testing)”)

powerbacgwas program: according to the wiki, I executed the following command in a computing cluster “nextflow run nextflow/main.nf --outprefix kpn_clade.var --tree ./data/kpn_clade.tree.annotated.nwk --vcf ./data/kpn_clade.vcf.gz --parameters ./data/kpn_clade.parameters.binary.efs.txt --run_phensim_vcf --num_cores 8 --gwas_tmp_dir tmp/gwas_tmp_kpn_clade_var_efs/”, which is taking more than a whole day and has not yet finished. I suggest to write usual computation time in the wiki or manuscript for users to understand

it in advance.

We have added more information on the Wiki page regarding computation time:

“If you have access to an LSF cluster, we recommend to run *PowerBacGWAS* Nextflow pipeline with the LSF executor for faster running times, wherein each process is submitted as a separate job by using the `bsub` command. This execution mode can be selected by adding the option ‘`-profile lsf`’ to the Nextflow command. You may need to edit the configuration file (`nextflow.config`) to create a new profile matching your system’s cluster configuration, see Nextflow’s configuration and executor for information on how to do this.”

And the following text to the manuscript (lines 563-571):

The computational time of running *PowerBacGWAS* depends on the number of parameter combinations, and number of variant sampling and phenotype simulation repetitions indicated in the parameters file. All these parameters combined determine the number of individual GWAS runs. We recommend to run the Nextflow pipeline with the LSF executor, if an LSF cluster is available, for faster running times, wherein each process is submitted as a separate job. Overall, for the six bacterial datasets used in this work, *PowerBacGWAS* used a median of 1,670 CPU hours (interquartile range: 548 to 3,542) and a median duration of 3.1 hours (interquartile range: 2 to 7.6 hours) when using the Nextflow LSF executor.